# PPM1D suppresses p53-dependent transactivation and cell death by inhibiting the Integrated Stress Response

Zdenek Andrysik[1,2] ✉, Kelly D. Sullivan[1,3], Jeffrey S. Kieft[4] & Joaquin M. Espinosa [1,2] ✉

The p53 transcription factor is a master regulator of cellular stress responses inhibited by repressors such as MDM2 and the phosphatase PPM1D. Activation of p53 with pharmacological inhibitors of its repressors is being tested in clinical trials for cancer therapy, but efficacy has been limited by poor induction of tumor cell death. We demonstrate that dual inhibition of MDM2 and PPM1D induces apoptosis in multiple cancer cell types via amplification of the p53 transcriptional program through the eIF2α-ATF4 pathway. PPM1D inhibition induces phosphorylation of eIF2α, ATF4 accumulation, and ATF4-dependent enhancement of p53-dependent transactivation upon MDM2 inhibition. Dual inhibition of p53 repressors depletes heme and induces HRI-dependent eIF2α phosphorylation. Pharmacological induction of eIF2α phosphorylation synergizes with MDM2 inhibition to induce cell death and halt tumor growth in mice. These results demonstrate that PPM1D inhibits both the p53 network and the integrated stress response controlled by eIF2α-ATF4, with clear therapeutic implications.

The key role of the transcription factor p53 in tumor suppression is documented by the high frequency of inactivating mutations in the *TP53* locus observed across diverse human cancers[1]. The p53 protein directly transactivates hundreds of target genes involved in numerous anti-tumoral responses including cell cycle arrest, apoptosis, DNA repair, and senescence[2,3]. p53 constitutes a major signaling hub in the cellular response to stress, being activated by a wide range of stimuli including DNA damage, oncogene activation, reactive oxygen species (ROS), and nutrient deprivation[4]. Through various mechanisms, stress signaling pathways attenuate the activity of key p53 repressors, most prominently MDM2 and MDM4. Both repressors inhibit p53 activity by obstructing its N-terminus transactivating domain, but only MDM2 promotes p53 degradation by the ubiquitin-dependent proteasome[5–8]. Another potent repressor of p53 is the protein phosphatase PPM1D (Protein Phosphatase,

Mg[2+]/Mn[2+] Dependent 1D, also known as WIP1, Wild Type p53-Induced Phosphatase 1)[9]. Both *MDM2* and *PPM1D* are direct p53 target genes, which creates negative feedback loops to control p53 activity[6,9]. Despite its well demonstrated role in control of p53 function, the mechanism of action of PPM1D is not well defined. It has been shown that PPM1D removes phosphate groups from both p53 and MDM2[10,11] and that it can also dephosphorylate key mediators of the DNA-damage response such as ATM[12] and CHK2[10]. Therefore, it has been proposed that whereas PPM1D acts solely to restore basal p53 activity following an activation event, MDM2 and MDM4 play an additional role by maintaining low levels of p53 activity in unstressed cells[13,14]. This notion is supported by experiments in mouse models, whereby depletion of either *Mdm2* or *Mdm4* causes embryonic lethality which can be rescued by concomitant loss of *Tp53*[15,16]. In contrast, *Ppm1d* knock-out mice are viable[17].

[1]Linda Crnic Institute for Down Syndrome, University of Colorado Anschutz Medical Campus, Aurora, CO 80045, USA. [2]Department of Pharmacology, University of Colorado Anschutz Medical Campus, Aurora, CO 80045, USA. [3]Department of Pediatrics, Section of Developmental Biology, University of Colorado Anschutz Medical Campus, Aurora, CO 80045, USA. [4]Department of Biochemistry and Molecular Genetics and RNA Bioscience Initiative, University of Colorado Anschutz Medical Campus, Aurora, CO 80045, USA. ✉e-mail: zdenek.andrysik@cuanschutz.edu; joaquin.espinosa@cuanschutz.edu

Given that nearly half of cancers express wild type p53, much effort has been devoted to the development of therapeutic strategies that could activate p53 to induce tumor regression. Several small molecules and peptides targeting MDM2 and/or MDM4 have been developed to activate p53 without the undesired effects of the genotoxic stress caused by conventional chemotherapy and radiation[18]. However, since development of the first-in-class MDM2 inhibitor nutlin-3a[19], it has become evident that although these compounds effectively activate p53 and its downstream transcriptional program, including induction of numerous pro-apoptotic genes, most cancer cell types undergo a reversible cell cycle arrest response of little therapeutic value[20]. Moreover, both in vitro experiments and results from clinical trials demonstrate the rapid selection of p53 mutant clones and development of drug resistance[2,21–24]. In clinical trials, use of these compounds associated with various adverse events, most prominently hematological toxicity (neutropenia and thrombocytopenia), as well as nausea/vomiting, asthenia, and diarrhea[25]. These results elicited many efforts to identify combinatorial therapies that could enhance the therapeutic potential of MDM2/MDM4 inhibitors[26,27]. Interestingly, shortly after a specific PPM1D inhibitor became available[28], a number of studies demonstrated synergistic effects of dual MDM2 and PPM1D inhibition resulting in an augmented apoptotic response in cancer cell lines both in vitro and in xenograft models[29–33]. However, the mechanisms driving this synergy remain unclear, as it was observed that p53 occupancy at target genes remained unchanged upon PPM1D inhibition[30], and that PPM1D inhibition also increased the apoptotic response to genotoxic drugs in p53 knock-out cells[30], thus suggesting the existence of additional p53-independent effects of PPM1D on promoting cell survival.

Here, to investigate the mechanism by which PPM1D blocks tumor cell death upon MDM2 inhibition, we employed papillary thyroid carcinoma cell lines expressing high levels of wild type PPM1D[34]. Notably, gain-of-function *PPM1D* mutations are frequently observed in thyroid carcinomas and other cancer types, often in a mutually exclusive fashion with *TP53* mutations, suggesting that PPM1D-mediated suppression of p53 activity is a common feature across diverse malignancies[2,35–40]. We identify the transcription factor ATF4 as a driver of the increased induction of p53 target genes and apoptosis observed upon dual inhibition of MDM2 and PPM1D. Furthermore, ATF4 is induced by the combinatorial treatment through the HRI-eIF2α aspect of the integrated stress response (ISR). These results reveal a key role for PPM1D in the cellular response to stress, whereby it not only inhibits the p53 network, but it also restrains the stress-induced alternative translation program elicited by inhibitory phosphorylation of the eIF2α complex. Moreover, we report a strong synergistic effect of combined pharmacological inhibition of MDM2 and eIF2α, resulting in a rapid apoptotic response in vitro and halted tumor growth and host survival in vivo. Given that translation rates are much higher in cancer cells compared to normal tissues[41], treatment strategies based on the combined activation of two stress response hubs−p53 and the ISR− represent a promising approach for treating p53 wild-type tumors.

## Results

### PPM1D inhibition increases p53-dependent transactivation upon MDM2 inhibition

It has been previously demonstrated that small molecule inhibitors of MDM2 and PPM1D synergize to elicit p53-dependent cell death in diverse cell types[29–33], and we have previously shown that gain-of-function PPM1D mutations are mutually exclusive with p53 mutations in thyroid carcinoma[2], suggesting that PPM1D restrains p53 activity in this cancer type. Therefore, we investigated the cellular response to MDM2 and PPM1D inhibition in two different thyroid carcinoma cell lines, TPC1 and K1. As seen in most cancer cell lines expressing wild type p53, MDM2 inhibition with nutlin stabilizes p53 but does not suffice to cause p53-dependent apoptosis in TPC1 or K1 cells (Fig. 1a, b

and Supplementary Fig. 1a). Inhibition of PPM1D catalytic activity with the small molecule inhibitor GSK2830371 does not stabilize p53 or induce apoptosis, but the combined inhibition of both p53 repressors elicits a clear apoptotic response along with increased phosphorylation of p53 on serine 15 within its N-terminus transactivation domain (Fig. 1a, b)[42]. As previously shown, p53 activation leads to PPM1D upregulation, but GSK2830371 treatment reduces PPM1D expression[28] (Supplementary Fig. 1b). In agreement with previous reports[29–32], the synergistic effect of the two inhibitors is also observed in other cancer cell lines, such as HCT116 (colorectal carcinoma), MCF7 (breast carcinoma), and SJSA (osteosarcoma) (Supplementary Fig. 1c). Furthermore, this synergy is also evident with more clinically-relevant MDM2 inhibitors such as idasanutlin (RG7388) and milademetan (Supplementary Fig. 1d). To investigate mechanisms of synergy upon dual inhibition of MDM2 and PPM1D, we completed transcriptome analysis of cells treated for 24 h with vehicle (DMSO), nutlin (10 μM), GSK2830371 (25 μM), or the combination of both drugs. Using DESeq2, we identified hundreds of mRNAs significantly upregulated or downregulated upon nutlin treatment (Fig. 1c and Supplementary Data 1). Notably, whereas PPM1D inhibition alone had little impact on the transcriptome, the combined treatment resulted in more differentially expressed genes than nutlin alone (Fig. 1c, d). Overlap analysis of genes significantly upregulated in each treatment ($q < 0.05$, fold change >1.5) showed that while many genes are upregulated by nutlin treatment with or without PPM1D inhibition, hundreds of genes reach statistical significance only when both p53 repressors are inhibited (Fig. 1d). Quantitative analysis revealed that most genes upregulated by nutlin treatment display greater fold increases upon concomitant inhibition of PPM1D, including most core direct p53 target genes[2], with clear gene-specific effects (Fig. 1e–g and Supplementary Fig. 1d). The increased transcriptional impact of p53 activation with the combined treatment is also reflected in the numbers of downregulated genes (Supplementary Fig. 1e, f). Ingenuity Pathway Analysis (IPA) demonstrated strong activation of the p53 transcriptional program upon MDM2 inhibition with or without PPM1D inhibition in both thyroid carcinoma cell lines (Supplementary Fig. 1g and Supplementary Data 2). Thus, although PPM1D inhibition on its own has little effect on gene expression, it enhances the output of the p53 transcriptional program. To investigate this further, we analyzed expression of canonical p53 target genes by Q-RT-PCR in additional cell lines, which confirmed the greater induction of multiple p53 targets in diverse cancer cell types (Fig. 1h and Supplementary Fig. 1h). To explore potential mechanisms driving this increased transactivation response for a considerable fraction of the p53 transcriptional program upon the combinatorial treatment, we performed an IPA upstream regulator analysis of the genes significantly upregulated during the combination treatment relative to MDM2 inhibition alone (Supplementary Fig. 1e). Interestingly, in both thyroid carcinoma cell lines, the top predicted upstream regulator of this gene set is ATF4 (Activating Transcription Factor 4, Fig. 1i and Supplementary Fig. 1i), leading to the hypothesis that activation of this transcription factor, a known mediator of the transcriptional program elicited by the ISR[43], could explain some of the differential effects observed upon dual inhibition of p53 repressors. Furthermore, ATF4 is the top predicted upstream regulator of the 52 genes induced by single PPM1D inhibition in the TPC1 cell line (Supplementary Fig. 1j).

Altogether, these results indicate that PPM1D restrains the p53 transcriptional program upon MDM2 inhibition through a mechanism likely involving the ATF4 transcription factor.

### Post-transcriptional activation of ATF4 drives the p53 response toward apoptosis

ATF4 belongs to a family of DNA-binding proteins that includes the AP-1 family of transcription factors, cAMP-response element binding proteins (CREBs) and CREB-like proteins[44]. Notably, the related factor ATF3

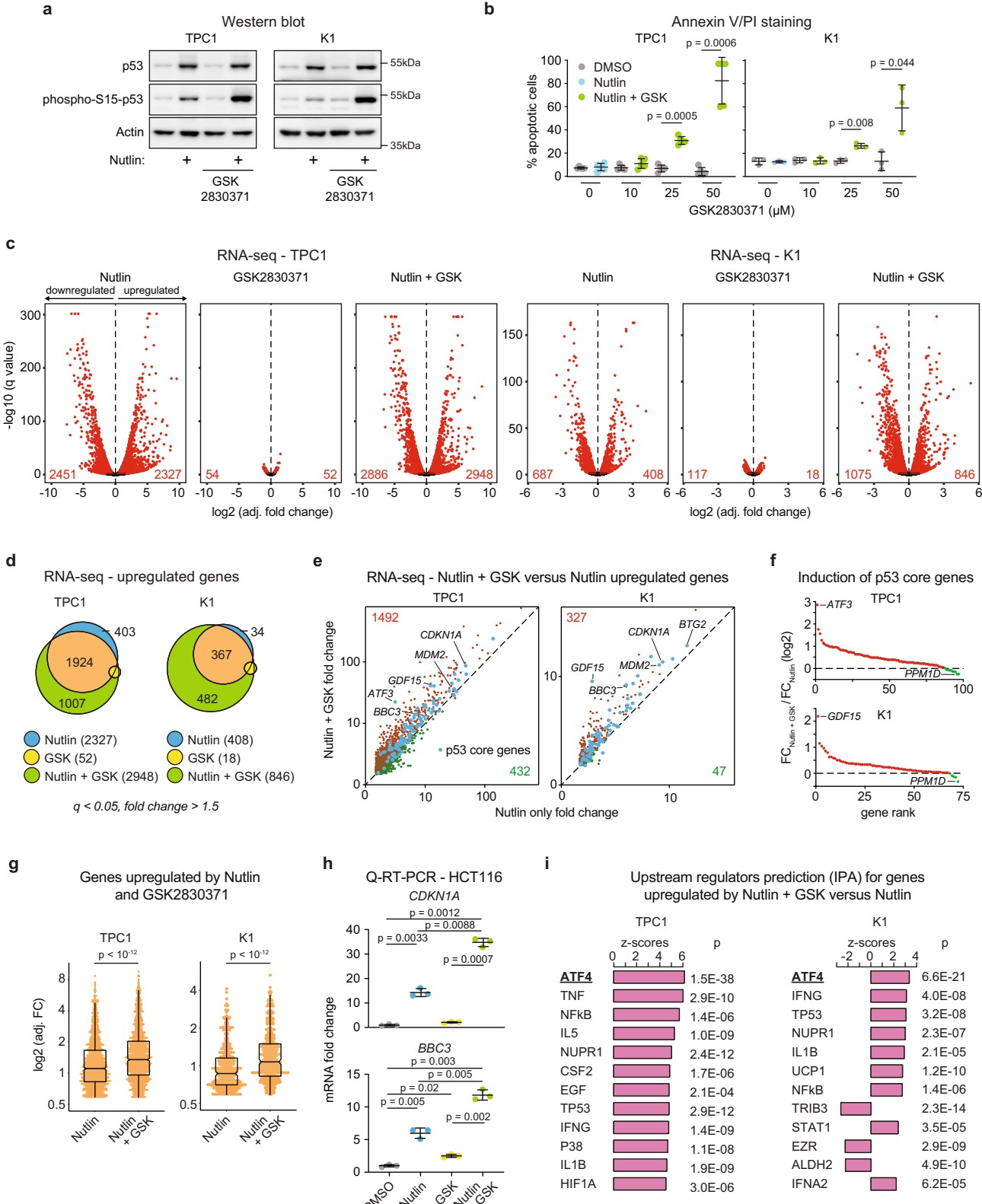

is a known direct target of p53[2,45,46], and ATF3 is also a target of ATF4[47]. Therefore, we investigated the regulation of *ATF3* and *ATF4* gene expression in our experimental paradigm. First, RNA-seq analysis revealed clear induction of *ATF3* but not *ATF4* at the mRNA level upon nutlin treatment in TPC1, K1, and four other cell lines investigated (Fig. 2a). This differential impact of p53 activation on ATF family members can be explained by the presence of a previously

characterized p53 enhancer upstream of the *ATF3* locus as seen by ChIP-seq[2,48], whereas no p53 binding is evident within 50 kb of the *ATF4* locus (Fig. 2b). Analysis of GRO-seq datasets at 60 min of nutlin treatment shows clearly rapid transactivation of *ATF3* but not *ATF4* in three different cell lines (Fig. 2c). *ATF4* mRNA analysis revealed only modest changes over the course of 48 h of nutlin treatment, being elevated at early time points and downregulated later on (Supplementary

**Fig. 1 | Dual inhibition of MDM2 and PPM1D potentiates the p53 transcriptional program. a** Western blots of TPC1 and K1 cells treated with vehicle (0.2% DMSO), 10 µM nutlin-3a, 25 µM GSK2830371, or both drugs for 24 h. Results shown here are representative of three independent experiments. **b** TPC1 and K1 cells were treated as indicated for 72 h. Cells were stained with Annexin V-FITC/propidium iodide (PI) and analyzed by flow cytometry. Data are represented as mean ± SD. Indicated statistical significance was calculated by paired, two-sided $t$ test, $n = 3$ independent experiments. **c** Differential expression analysis of RNA-seq reads in cells treated 24 h with 10 µM nutlin-3a, 25 µM GSK2830371, or both drugs compared to vehicle-treated controls. Red points and numbers indicate significantly upregulated and downregulated genes (DESeq2, $q < 0.05$, adjusted fold change >1.5). **d** Overlaps among indicated groups of significantly upregulated genes identified in **c**. **e** Comparison of relative fold induction in cells treated with nutlin alone versus the drug combination. Red points indicate genes with a fold change upon combination treatment greater than the fold change with nutlin alone. Green points denote a fold change upon combination treatment lower than the fold change with nutlin alone. Blue points indicate previously identified direct core p53 target genes. Log10 scale is used for both plots. **f** Changes in induction of direct core p53 target genes in cell populations treated with nutlin alone versus the drug combination. Box plots center lines represent median values, box boundaries outline the 25th and 75th percentile. Whiskers depict the smallest or largest values within 1.5 times of the interquartile range. **g** Adjusted fold changes of genes induced ($q < 0.05$, adjusted fold change >1.5) by both nutlin and nutlin + GSK2830371 in TPC1 ($n = 1924$ genes) and K1 ($n = 367$ genes). The indicated $p$ value has been calculated with Wilcoxon signed-rank test. **h** Q-RT-PCR of *CDKN1A* and *BBC3* in HCT116 cells treated for 24 h with indicated compounds. Data are represented as mean ± SD. Paired, two-sided t test has been used to calculate the indicated $p$ value, $n = 3$ independent experiments. **i** Ingenuity Pathway Analysis of upstream regulators in genes upregulated by the drug combination significantly more ($q < 0.05$, adjusted fold change >1.5) than by nutlin alone. Indicated $p$ values were calculated with Fisher's Exact test without correcting for multiple comparisons. See also Supplementary Fig. 1. Source data are provided as a Source data file.

Fig. 2a, b). Second, *ATF3* mRNA expression is further increased upon concurrent PPM1D inhibition as seen by RNA-seq in TPC1 and K1 cells (Fig. 2a) and by Q-RT-PCR in HCT116 cells (Supplementary Fig. 2a). Third, we observed clear induction of both ATF3 and ATF4 at the protein level upon dual inhibition of the p53 repressors (Fig. 2d and Supplementary Fig. 2c, d). Whereas ATF3 protein accumulation can be explained by induced mRNA expression, ATF4 protein induction is likely due to post-transcriptional control. In fact, it has been well demonstrated that ATF4 translation can be increased in some cellular settings[49]. Thus, the increased levels of ATF4 protein could explain the synergistic effect of the drug combination on ATF3 expression, as *ATF3* is a transcriptional target of both p53 and ATF4[45–47].

Next, we tested the contribution of ATF3 and ATF4 to the apoptotic response and induction of p53 target genes upon dual inhibition of MDM2 and PPM1D. Indeed, knockdown of either transcription factor significantly reduced the number of apoptotic cells after combinatorial treatment with nutlin and GSK2830371 (Fig. 2e, f and Supplementary Fig. 2e, f). We then tested the effects of ATF4 overexpression using a stable integrated, doxycycline-inducible vector (Supplementary Fig. 2g). Whereas ATF4 overexpression had no significant effect on its own or in cells treated with nutlin or GSK2830371 alone, it further increased the apoptotic signal observed during the combinatorial treatment (Fig. 2g). Lastly, Q-RT-PCR analysis showed that ATF4 knockdown decreases, and ATF4 overexpression increases, expression of multiple p53 target genes (Fig. 2h–i and Supplementary Fig. 3a, b). Knockdown of ATF3 also reduced induced expression of several p53 target genes that require ATF4 (Supplementary Fig. 3c). Analysis of available ATF4 ChIP-seq data demonstrated that ATF4 binds to ~38% and ~60% of the genes more strongly induced by the combinatorial treatment in K1 and TPC1 cell lines, respectively, but only to ~8% and 18% of those who are not further induced by PPM1D inhibition (Supplementary Fig. 3d). Moreover, ChIP q-PCR suggests increased occupancy of the ATF4 transcription factor at all tested p53 target genes (Fig. 2j). Lastly, we investigated whether dual inhibition of MDM2 and PPM1D would induce *DDIT3* (encoding CHOP), another ATF4 target gene with prominent roles in apoptosis[50]. Indeed, CHOP was induced at the protein level by the combinatorial drug treatment (Supplementary Fig. 3e) and its knockdown reduced the apoptotic response (Supplementary Fig. 3f).

Altogether, these results demonstrate that dual inhibition of MDM2 and PPM1D induces the ATF4 pathway, which in turn contributes to greater transactivation of some p53 target genes and p53-dependent apoptosis.

## ATF4 stabilization downstream of the HRI-eIF2α axis upon dual inhibition of p53 repressors

ATF4 protein expression is tightly controlled at the translational level downstream of the ISR[51]. After diverse stress stimuli, the ISR signaling cascade shuts down most mRNA translation through inactivating phosphorylation of the eIF2α translation factor[52]. However, these events lead to increased selective translation of ATF4 and other mRNAs through a bypass mechanism involving upstream open reading frames (uORFs) and non-canonical initiation factors[53,54]. Four major protein kinases can induce eIF2α phosphorylation in response to diverse stimuli, including HRI (Heme-Regulated Inhibitor, encoded by *EIF2AK1*), PKR (Protein Kinase R, encoded by *EIF2AK2*), PERK (PKR-like Endoplasmic Reticulum Kinase, encoded by *EIF2AK3*), and GCN2 (General Control Nonderepressible 2, encoded by *EIF2AK4*) (Fig. 3a). Therefore, we tested the impact of MDM2/PPM1D inhibition on eIF2α signaling. Indeed, dual inhibition of the p53 repressors led to increased eIF2α phosphorylation at residue serine 51 (S51) in multiple cell lines (Fig. 3b), which prompted us to analyze global effects on translation upon each treatment using polysome profiling analysis. Notably, inhibition of either MDM2 or PPM1D caused decreases in the polysome/monosome ratio, but the repressive effect on translation was much greater with the combinatorial treatment (Fig. 3c, d). Expectedly, the ATF4 mRNA shifted toward heavier polysomal fractions with the combinatorial treatment, whereas the control GAPDH mRNA shifted toward lighter fractions, indicative of selective ATF4 mRNA translation concurrent with global translational repression in this setting[49] (Supplementary Fig. 4a). Next, we investigated upstream signaling events, which revealed elevation of HRI protein levels, but not so for PKR, PERK or GCN2, upon dual inhibition of p53 repressors in multiple cell lines (Fig. 3e and Supplementary Fig. 4b). Protein levels of HRI (encoded by *EIF2AK1*) increased despite downregulation of its mRNA and without clear changes in its polysomal distribution, potentially indicative of protein stabilization (Supplementary Fig. 4c, d) as reported previously[55]. Notably, knockdown of HRI (Supplementary Fig. 4c, e) blocked both eIF2α phosphorylation and ATF4 induction upon the combinatorial treatment with nutlin and GSK2830371 (Fig. 3f) and the downstream apoptotic response (Fig. 3g). In contrast, knockdown of PKR, PERK, or GCN2 did not reduce apoptosis in response to the combinatorial treatment (Supplementary Fig. 4f, g).

To define the mechanism by which dual inhibition of the p53 repressors activates HRI, we tested each of the main known stimuli leading to HRI induction, including decreased proteasomal activity[55,56], mitochondrial collapse[57], increase in reactive oxygen species (ROS)[58], and depletion of cellular heme[59]. Inhibition of p53 repressors, either individually or in combination, did not have significant effects on either total proteasomal activity (Supplementary Fig. 5a, b) or mitochondrial membrane potential collapse at early time points (24 h), before the onset of apoptosis at the time of HRI induction (Supplementary Fig. 5c, d). However, dual inhibition of p53 repressors caused a modest but significant increase in ROS (Supplementary Fig. 5e, f), and, most prominently, a strong depletion of cellular regulatory (free) heme in multiple cell lines as soon as 6 h post-treatment (Fig. 3h and

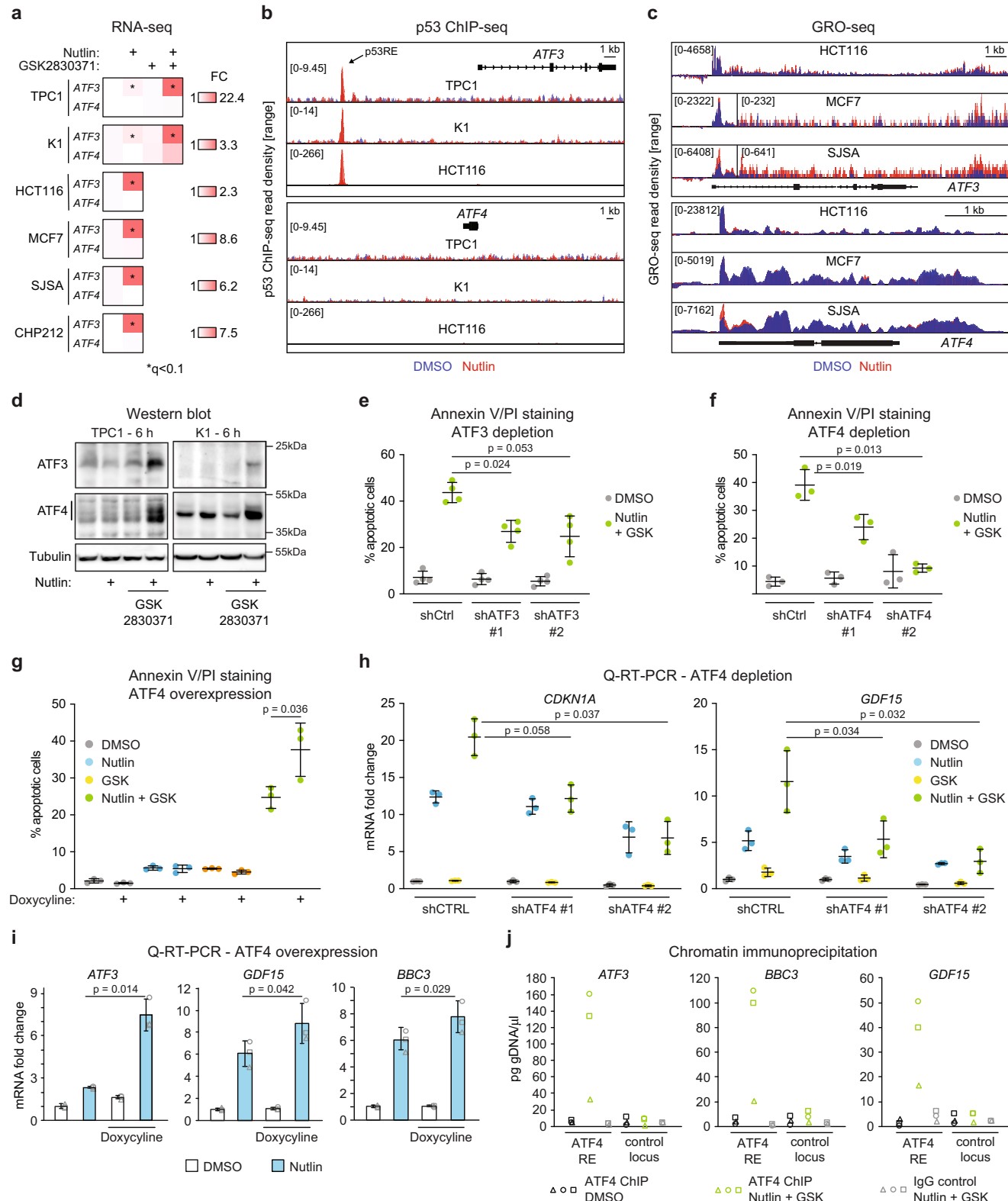

Supplementary Fig. 5g). Notably, heme metabolization can lead to elevated ROS levels[60], suggesting that the upstream event triggered by the combinatorial treatment could be degradation of heme. To investigate this further, we tested for changes in expression of HMOX1 (heme oxygenase 1, HO-1), the inducible isoform of the rate-limiting enzyme of heme degradation[61]. Indeed, HMOX1 protein expression was synergistically induced upon dual inhibition of p53 repressors (Fig. 3i). Furthermore, HMOX1 knockdown led to accumulation of

regulatory heme and reduced the apoptotic response upon dual inhibition of MDM2 and PPM1D (Fig. 3j–k and Supplementary Fig. 5h). This indicates that heme depletion by HMOX1 may be the initiating event, leading to heme depletion and downstream Fe-induced ROS elevation (Supplementary Fig. 5i), a notion supported by the fact that increased ROS levels were not reduced by treatment with N-acetyl cystine (NAC), an antioxidant that cannot prevent $Fe^{2+}$-induced ROS elevation[62] (Supplementary Fig. 5e, f and Fig. 3l).

**Fig. 2 | ATF4 is required for the apoptotic response upon dual inhibition of MDM2 and PPM1D. a** *ATF3* and *ATF4* mRNA induction in cell lines treated with vehicle (0.2% DMSO), 10 μM nutlin-3a, 25 μM GSK2830371, or both drugs for 24 h. *q* values were calculated by the DESeq2 software. **b** p53 occupancy at *ATF3* and *ATF4* gene loci analyzed by ChIP-seq. **c** Transcriptional activity at *ATF3* and *ATF4* gene loci measured by GRO-seq in cell lines treated with 10 μM nutlin-3a for 60 min. **d** Western blots in TPC1 and K1 cell lines treated with indicated compounds for 6 h. Results shown here are representative of three independent experiments. **e**, **f** TPC1 cells depleted of ATF3 or ATF4 were treated for 72 h, stained with Annexin V-FITC/PI and analyzed by flow cytometry. Statistically significant difference was calculated by paired, two-sided *t* test, *n* = 4 (**e**), *n* = 3 (**f**) independent experiments.

**g** TPC1 cells transduced with Tet-on-*ATF4* expression vector were treated with 10 μg/ml doxycycline, vehicle control, or drug combination for 72 h prior to flow cytometry measurement of Annexin V-FITC/PI positive cells. Paired, two-sided *t* test was used for calculations of statistical significance (*n* = 3 independent experiments). **h**, **i** Q-RT-PCR of p53 target genes in the TPC1 cell line treated with the indicated compounds for 24 h. Paired, two-sided *t* test was used to calculate the indicated *p* value, *n* = 3 independent experiments. Data in **e**–**i** are represented as mean ± SD. **j** ATF4 occupancy analyzed by chromatin immunoprecipitation (ChIP) followed by Q-PCR at p53 target genes. See also Supplementary Figs. 2 and 3. Source data are provided as a Source data file.

Altogether, these results illuminate a mechanism by which dual inhibition of MDM2 and PPM1D induce ATF4 activity to convert the cellular response to p53 from cell cycle arrest to cell death.

## Pharmacological inhibition of eIF2α synergizes with MDM2 inhibition to elicit apoptosis

Next, to further investigate the role of the ISR in control of the p53 response, we tested the hypothesis that pharmacological inhibition of eIF2α could synergize with nutlin to elicit p53-dependent apoptosis. Toward this end, we employed small molecule inhibitors of the protein phosphatase 1 (PP1) complex, a major phosphatase in the control of eIF2α activity[63] (Fig. 4a). We fist used nelfinavir, a compound that induces the ISR by downregulating the PP1 cofactor CReP (Constitutive Repressor of eIF2α Phosphorylation, PPP1R15B)[64]. Nelfinavir treatment alone induced eIF2α phosphorylation (Fig. 4b) and decreased the polysome/monosome ratio (Fig. 4c, d). Although treatment with nelfinavir alone stabilized ATF4, it did not suffice to induced caspase 3 cleavage (Fig. 4e). However, in combination with nutlin, nelfinavir increased ATF3 expression, induced strong caspase 3 activation, and apoptosis in diverse cell types with strong synergy (Fig. 4e, f and Supplementary Fig. 6a–c) as well as complementary MTT assays of cellular metabolic activity (Fig. 4g and Supplementary Fig. 6d). Similar results were obtained with sal003, a structurally diverse small molecule inhibitor of the PP1 complex acting via repression of the regulatory subunit PPP1R15A (GADD34)[65] (Fig. 4f, g and Supplementary Fig. 6d).

These results suggest that p53-dependent apoptosis, which is known to require transactivation of key pro-apoptotic genes, somehow bypasses translational inhibition triggered by the ISR. To test this, we analyzed the distribution of the mRNA encoding for PUMA/BBC3, a key mediator of p53-dependent apoptosis[66,67], in polysomal fractions. Indeed, PUMA mRNA was enriched in heavy (i.e., highly translated) polysomal fractions under conditions of ISR activation (Fig. 4h), along with increased PUMA protein levels (Fig. 4i).

Altogether, these results point to a functional crosstalk between two major stress responses, the transcriptional program controlled by p53, and the translational response governed by eIF2α and ATF4, with major impacts on control of cell viability.

## Dual inhibition of MDM2 and eIF2α exerts synergistic anti-tumoral activity

Next, we performed a pre-clinical test of the synergistic effects of MDM2 inhibition and nelfinavir. An analysis of gene expression for various components of the eIF2α translational complex in diverse cancer types revealed consistent and statistically significant upregulation of multiple subunits in colon adenocarcinomas (COAD) (Fig. 5a). Using the HCT116 COAD cell line, we observed synergistic induction of apoptosis when nelfinavir was combined with three structurally different MDM2 inhibitors, including milademetan, a compound which is currently being tested in Phase III clinical trials (Fig. 5b)[68]. The synergistic apoptotic effect of MDM2 inhibition and nelfinavir was also observed in three-dimensional COAD organoids (Fig. 5c). This prompted us to test the combinatorial treatment in a COAD xenograft

model using HCT116 cells. HCT116 cells were injected in the flanks of nude mice to establish tumors and after two weeks of tumor engraftment, mice were treated with the MDM2 inhibitor milademetan, nelfinavir, or both drugs in combination (Supplementary Fig. 7a, b). Tumors continued to grow in the vehicle-treated mice as well as in those treated with each drug individually. However, the combinatorial treatment had a significant effect on tumor growth (Fig. 5d, e). Kaplan–Meier analysis revealed that all animals treated with vehicle or single treatments had to be sacrificed at the humane endpoint (tumor size >1000 mm³), whereas all mice receiving the combinatorial treatment survived up to 4 weeks of treatment (Fig. 5f). Notably, despite its strong anti-tumoral activity, the combinatorial treatment did not significantly impact animal body weight (Supplementary Fig. 7c). Q-RT-PCR analysis confirmed that the combinatorial treatment led to stronger induction of p53 target genes, including *ATF3*, in tumors (Fig. 5g and Supplementary Fig. 7d). Histology analysis of the tumors confirmed that the combinatorial treatment had a more significant effect on cell proliferation than each drug alone (Fig. 5h, i).

Altogether, these results demonstrate a pharmacological strategy to enhance the anti-tumoral activity of p53 upon MDM2 inhibition.

## Discussion

Despite many efforts to develop targeted drugs that could restore p53 function, either through reactivation of mutant p53 or inhibition of p53 repressors[18], p53-based therapies remain an unfulfilled promise in modern cancer treatment. Most cancer cell types expressing wild type p53 undergo reversible cell cycle arrest upon non-genotoxic p53 activation, with a p53-dependent apoptotic response being observed only in a small fraction of the cellular population or in a handful of very sensitive cell lines, which clearly limits the therapeutic potential of these agents. Moreover, similarly to other targeted cancer therapeutics, prolonged use of MDM2 inhibitors leads to development of resistance, mostly through selection of mutant p53 cell clones[2,21–24]. Therefore, it is important to identify mechanisms restraining the anti-tumoral effects of p53 during pharmacological reactivation in the clinic. One promising avenue is the identification of druggable targets within pathway(s) shielding cells from p53-driven apoptosis, which could in turn enable the design of efficient combinatorial cancer therapies.

Within this context, the observation that dual inhibition of mechanistically distinct p53 repressors switches the cellular response to p53 activation from cell cycle arrest to apoptosis merits further investigation. On their own, MDM2 inhibitors (e.g., nutlin, idasanutlin, siremadlin/HDM201, milademetan) and the PPM1D inhibitor GSK2830371 show limited effects on cell viability, but dual inhibition of the p53 repressors provokes an apoptotic response in cancer cell types of diverse origin[29,31,32,69]. However, further development of this promising treatment strategy has been hampered by modest insight into the underlying mechanism. As p53-mediated transactivation is instrumental to the onset of apoptosis triggered by combinatorial MDM2/PPM1D inhibition[31,33,70], we embarked on a genome-wide investigation of changes in the p53 transcriptional program upon single versus dual inhibition of MDM2 and PPM1D. Interestingly, dual inhibition of the

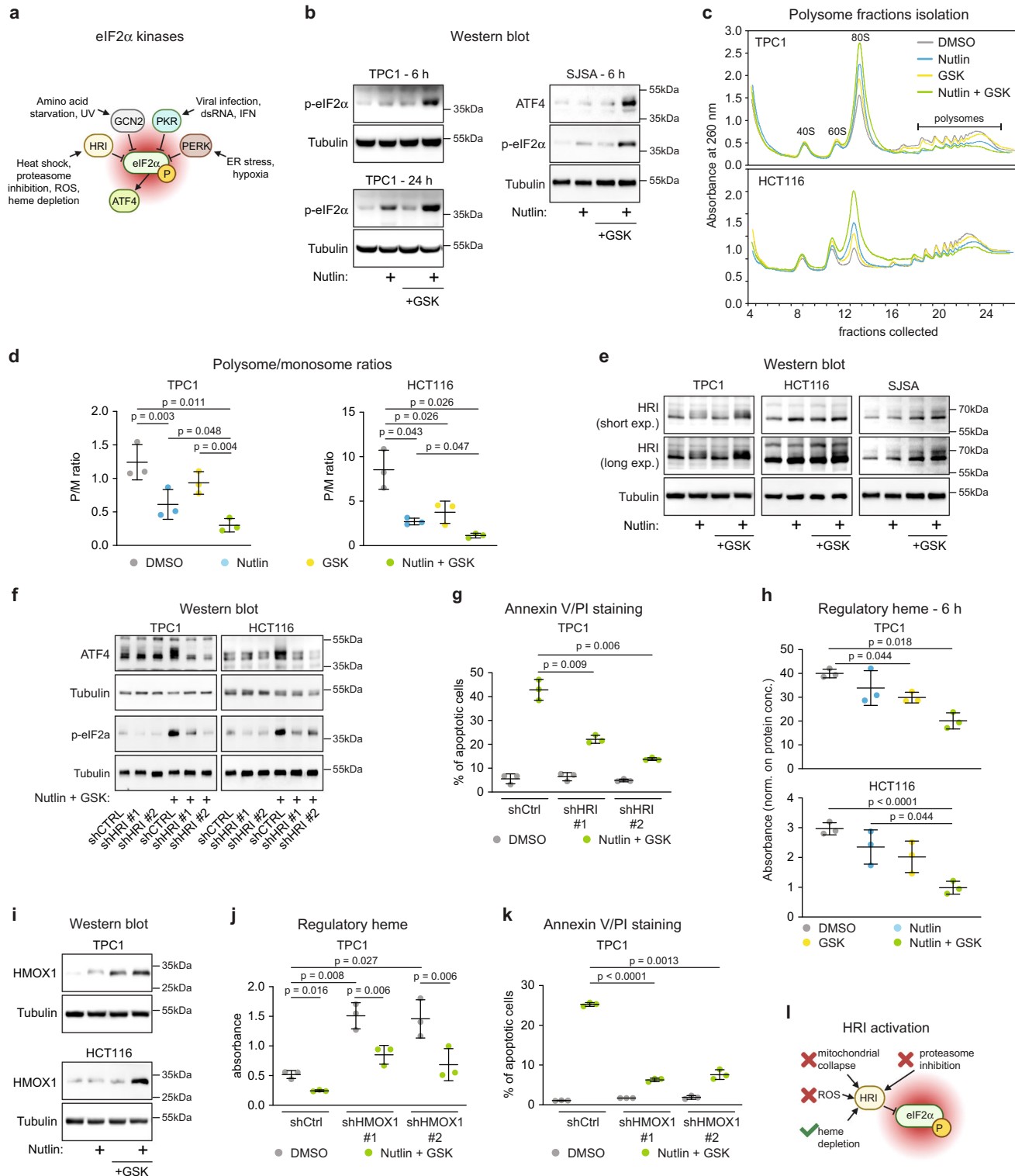

p53 repressors led to a clear amplification of the p53 transcriptional program, both in numbers of genes significantly upregulated and the magnitude of changes observed. Notably, in experimental systems using low doses of MDM2 inhibitors or DNA-damaging drugs resulting in only partial disruption of the MDM2-p53 interaction, dual use of MDM2 and PPM1D inhibitors led to increased total p53 levels[29,31,69]. However, in our experimental paradigm we observed increased transcriptional output even though p53 levels were similar upon single or dual inhibition of its repressors, which prompted us to investigate the mechanism by which PPM1D inhibition would boost the p53

transcriptional program, leading to identification of the AP-1 transcription factor family member ATF4 as the key mediator of these effects.

A functional interplay between p53 and AP-1 family members has been documented in diverse settings[71]. Notably, the AP-1 family member *ATF3* is a direct target gene of both p53[45,46] and ATF4[47]. Moreover, ATF3 and ATF4 share their binding partners within the AP-1 family, sequence specificity, as well as a role of transcriptional cofactors of p53[72–75]. Given the high occurrence of AP-1 sites across the genome, including at most open chromatin sites, AP-1 family members

**Fig. 3 | ATF4 accumulation upon dual MDM2/PPM1D downstream of heme depletion, HRI induction, and eIF2α phosphorylation. a** Schematic of signaling pathways leading to inhibitory phosphorylation of eIF2α. The key residue mediating this inhibitory phosphorylation is serine 51. Created with BioRender.com. **b** Western blots of cells treated with vehicle (0.2% DMSO), 10 μM nutlin-3a, 25 μM GSK2830371, or the drug combination for indicated times. p-eIF2α indicates S51 phosphorylation. **c** Cells treated with indicated compounds for 24 h were lysed and subjected to polysome profiling by using sucrose density gradient fractionation. **d** Polysome to monosome ratios. Absorbances displayed in **c** were quantified, and statistical significance ($n = 3$ independent experiments) was calculated using paired, two-sided t test. **e** Western blots in cells treated with indicated compounds for 24 h. **f** Western blots in cells transduced with non-targeting shRNA controls (shCTRL) or two different shRNAs targeting HRI (*EIF2AK1*). p-eIF2α indicates S51 phosphorylation. **g, k** TPC1 cells depleted of HRI (**g**) or HMOX1 (**k**) were treated with vehicle control or the drug combination for 72 h. Fraction of apoptotic cells was determined by flow cytometry. Statistical significance ($n = 3$ independent experiments) was calculated by paired, two-sided t test. **h, j** Cellular levels of free (regulatory) heme were measured in cells treated with denoted compounds for 6 h. Paired, two-sided t test was used for calculations of statistical significance ($n = 3$ independent experiments). **i** Western blots in cells treated with the indicated compounds for 24 h. **l** A schematic of HRI activation upon dual inhibition of MDM2 and PPM1D. Created with BioRender.com. Results shown in **b, e, f, i** are representative of three independent experiments. Data in **d, g, h, j, k** are represented as mean ± SD. See also Supplementary Figs. 4 and 5. Source data are provided as a Source data file.

converge promiscuously on enhancers to potentiate transcription[43,74]. Because both ATF3 and ATF4 were induced by the combination of MDM2/PPM1D inhibitors and were similarly required for the apoptotic response, we focused on elucidating the mechanism of activation of ATF4, which acts upstream of ATF3. Our results indicate that ATF4 induction is associated with inhibitory phosphorylation of the eIF2α subunit of the eIF2 translation initiation factor, a well-established mechanism of ATF4 protein upregulation by selective translation from uORFs[53].

Inhibitory phosphorylation of eIF2α at serine 51 by diverse upstream kinases integrates the cellular response to a broad suite of stress stimuli to promote cell survival[76,77]. Notably, previous work reported upregulation of the eIF2α kinase PKR (encoded by *EIF2AK2*) at the mRNA level downstream of p53 activation, leading to eIF2α phosphorylation and ATF4 induction[78]. However, our analysis of dozens of -omics datasets found neither p53 binding sites at the *EIF2AK2* locus or transactivation of the gene upon p53 activation in multiple cell types examined[2]. Moreover, the *EIF2AK2* gene is not commonly upregulated by p53-activating stimuli[3]. In contrast, our results document a role for HRI in eIF2α phosphorylation and ATF4 induction in our experimental paradigm. The observed induction of HRI was accompanied by increased HMOX1 expression and decreased heme levels, along with increased cellular concentration of $Fe^{2+}$, which in turn could potentially trigger ferroptosis[79–81]. However, ferroptosis is an unlikely cause of cell death in our experiments, as the observed activation of caspase 3 in cells exposed to the combinatorial treatment is clearly indicative of apoptosis[29,69], and an exclusionary criteria for ferroptosis[80]. Notably, elevated intracellular concentration of $Fe^{2+}$ ions increases multiple types of cell death, including apoptosis via elevated ROS production through the Fenton reaction[82]. Since pre-treatment with NAC failed to protect cells from the ROS increase seen in our experiments as previously reported for $Fe^{2+}$-induced oxidative stress[62], $Fe^{2+}$ buildup as an outcome of heme degradation is the most likely cause of increased ROS levels in our system. Consistently, elevated ROS production has been described as mechanism that can convert the cellular response to p53 activation from cell cycle arrest to apoptosis[83–85]. Future studies would be needed to elucidate the mechanism leading to HMOX1 upregulation during combinatorial inhibition of p53 repressors, and ATF4 itself may be involved through a positive feedback loop, as ATF4 has been shown to transactivate HMOX in some settings[86].

Importantly, these results illuminate combinatorial pharmacological strategies to enhance p53-dependent tumor suppression via induction of the ISR with FDA-approved drugs such as nelfinavir and sal003. Nelfinavir, which inhibits HIV1 and HIV2 proteases, was approved for HIV treatment in 1997 as a safe and orally available drug[87]. However, it was later discovered that nelfinavir also represses the PP1 cofactor CReP to trigger a robust ISR without activation of eIF2α kinases[64]. Sal003 increases eIF2α phosphorylation status via repression of the regulatory subunit PPP1R15A (GADD34) of the PP1 complex[65]. Sal003 is a more potent and soluble derivative of

Salubrinal[88], a molecule that was shown to block eIF2α dephosphorylation mediated by the herpes simples virus and inhibit viral replication[65]. In cancer cells, the translational machinery makes up a large fraction of the cellular proteome[89], and it is strongly upregulated to fuel tumor growth[41,89,90], which provides the rationale for targeting translation initiation in cancer treatment. In fact, nelfinavir has shown tumor suppressive effects[87,91,92] and is being tested in numerous clinical trials. In combination with radiation and chemotherapy, nelfinavir showed promising results for treatment of pancreatic cancer[93], multiple myeloma, non-small cell lung cancer, colorectal cancer, and glioblastoma multiforme[94]. To our knowledge, targeted inhibition of eIF2α in combination with MDM2 inhibition has not been tested, and our preclinical test of the synergistic effects of milademetan and nelfinavir warrants further investigation. In support of this notion, previous work suggested that apoptosis following treatment with p53-reactivating drugs requires activation of the ISR[95,96].

In sum, our results illuminate a mechanism by which the PPM1D phosphatase coordinately opposes two major stress signaling pathways, the p53 network and the IRS, to promote the survival of cancer cells, with clear implications for the development of p53 reactivation strategies in the clinic.

## Methods

### Cell culture

TPC1, K1 (a derivative of GLAG-66), HCT116, MCF7, SJSA, and HEK293FT cells were cultivated in RPMI (TPC1, SJSA), DMEM (K1, MCF7, HEK293FT), and McCoy's (HCT116) media (Gibco, Thermo Fisher Scientific) supplemented with 10% fetal bovine serum (Peak Serum) and 1% antibiotic-antimycotic mixture (Gibco). Cells were plated a day before the treatment and maintained in a humidified atmosphere with 5% of $CO_2$ at 37 °C. Both TPC1 and K1 lines were gifts from Dr. Rebecca Schweppe, University of Colorado Anschutz Medical Campus (CU-AMC). Cell line identity has been verified by short tandem repeats profiling at the Cell Technologies Shared Resource, CU-AMC. Colorectal cancer organoid cultures (CRC172) were obtained from the Enteroid Stem Cell Core facility at CU-AMC.

Cell lines depleted of RNAi targets were prepared from the parental lines using lentiviral transduction. Briefly, HEK293FT cells were transfected with a mixture of shRNAs vectors (pLKO.1-puro/pLKO.5-puro, obtained from the Functional Genomics Facility at the CU-AMC) and packaging vector mix (pΔ8.9 and pCMV-VSV-G). Live lentiviral particles released into the cultivation media were sterile-filtered and combined with destination cell line cultures. After 48 h of puromycin selection at 10 μg/ml, surviving cells were expanded for experimental needs while any prolonged cultivation was avoided.

### Xenograft tumor model

Athymic nude mice (NU/NU) weighting 20–30 g and being 8–12 weeks of age (Charles River Labs) were housed in cages under standard conditions (22 °C, 50% relative humidity, 12-h light/dark cycles) and provided with food and water ad libitum. Next, $10^6$ exponentially growing

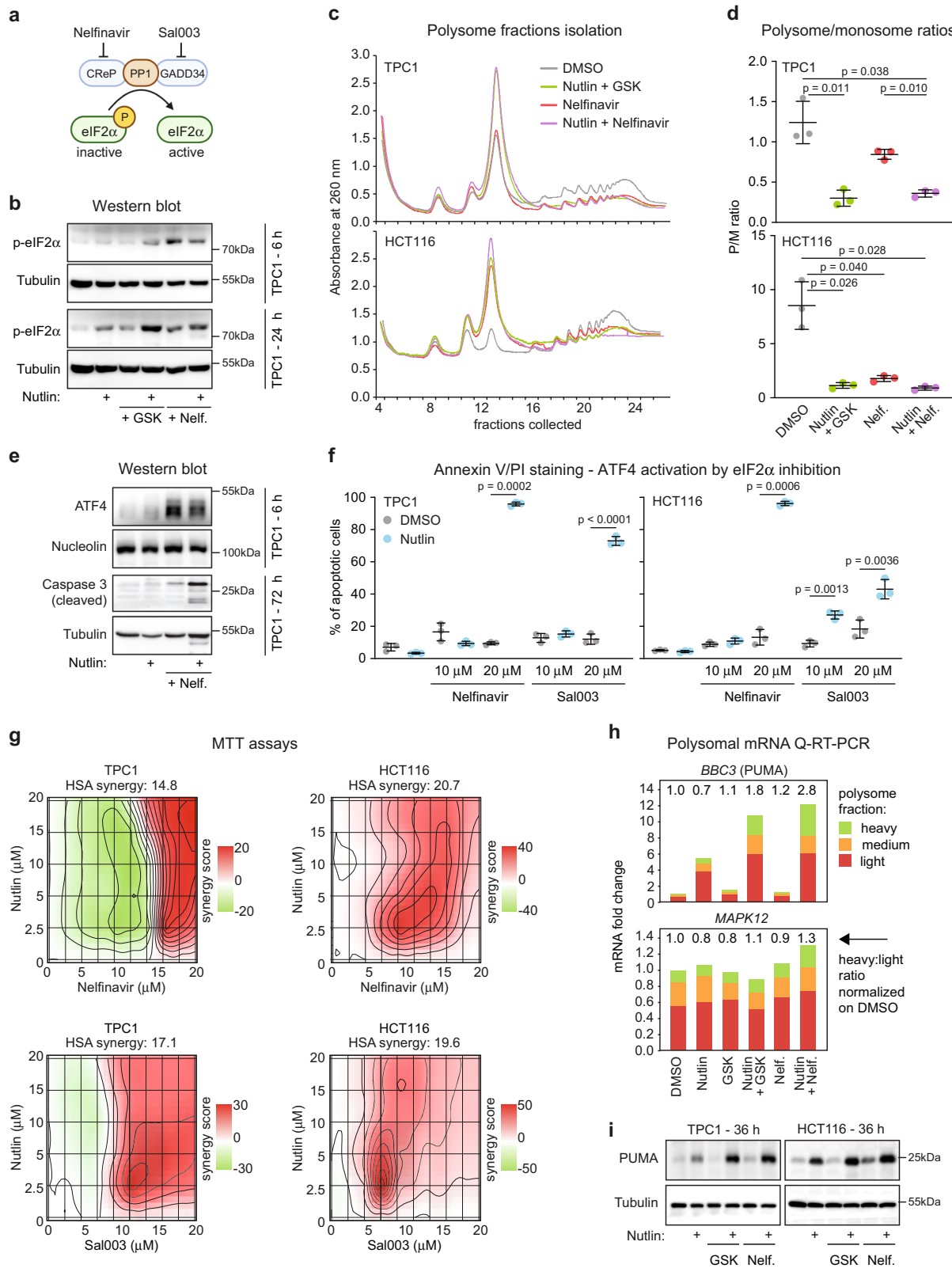

HCT116 cells resuspended in 100 µl Matrigel/PBS (5 mg/ml final) were injected subcutaneously into both flanks. Tumors grew for 10–14 days before treatment initiation. Experimental animals were given 200 mg/kg nelfinavir and/or 200 mg/kg milademetan (Rain Therapeutics) by oral gavage once daily, 5 days a week. Tested compounds were prepared in a mixture of 2% Klucel (hydroxypropyl cellulose), 0.5% Tween 80, and 35% ethanol. Tumor volumes (v) were estimated daily, 5 days a week using caliper measurements and formula $v = (l \times w^2)/2$, where $l$ represents the greatest length of the tumor and $w$ is tumor width in the perpendicular axis. Average volumes of the right and left flank tumors were used for plotting and calculations. Equal ratios of male and female mice were used in treatment groups (5 males and 5 females). All in vivo experiments were approved by the Institutional Animal Care and Use Committee at the CU-AMC (IACUC protocol 00432).

**Fig. 4 | Pharmacological inhibition of eIF2α synergizes with nutlin to induce cell death. a** Schematic of eIF2α inhibition by nelfinavir and sal003. Created with BioRender.com. **b** Western blots of cells treated with vehicle (0.2% DMSO), 10 μM nutlin-3a, 25 μM GSK2830371 (GSK), 20 μM nelfinavir (Nelf.), and drug combinations for indicated times. p-eIF2α indicates S51 phosphorylation. **c** Representative polysome profiles of cells treated with indicated compounds for 24 h. **d** Quantification of polysome to monosome ratio in polysome profiles from **c** (area under the curve, $n = 3$ independent experiments). Data are represented as mean ± SD. Statistical significance was calculated using paired, two-sided $t$ test. **e** Western blots in TPC1 cells treated as indicated. **f** TPC1 and HCT116 cells were treated with indicated compounds for 48 h. Fraction of apoptotic cells was determined by flow cytometry. Data are represented as mean ± SD. Paired, two-sided $t$ test was used for calculations of statistical significance ($n = 3$ independent experiments). **g** Absorbance values from using MTT assays were analyzed with SynergyFinder[107]. The degree of combination synergy was calculated using highest single agent (HSA) reference model. Synergy of specific concentrations (synergy score) was plotted to show synergy distribution. The higher the score, the stronger the synergy at those concentrations. **h** Q-RT-PCR of RNA isolated from polysome fractions of TPC1 cells as shown in **c** and in Fig. 3c. Light polysome samples were prepared with fractions 19-21, medium with fractions 22-23, and RNA associated with heavy polysomes was isolated from fractions 24 and 25. **i** Western blots in TPC1 and HCT116 cells treated as indicated for 36 h. See also Supplementary Fig. 6. Results shown in **b**, **e**, **i** are representative of three independent experiments. Source data are provided as a Source data file.

## Western blot

Protein samples were prepared with cells washed twice with PBS and lysed in modified Laemmli buffer[97] (1% w/v SDS, 10% w/v glycerol, 100 mM Tris pH 7.2, protease (cOmplete Mini, Roche) and phosphatase inhibitors (PhosSTOP, Roche). Following a brief sonication (2.5 W, 5 s) and heat denaturation (90 °C, 5 min), total protein concentration in whole-cell lysates was measured by a BCA Protein Assay Kit (Pierce, Thermo Fisher Scientific). Twenty micrograms of protein per sample was resolved by SDS-PAGE and transferred onto a 0.45 μm PVDF membrane (0.2 μm for ATF3 detection). Membranes were incubated in blocking buffer (5% fat-free milk in wash buffer−20 mM Tris pH 7.6, 150 mM NaCl, 0.2% Tween 20) for an hour at RT and o/n at 4 °C with the primary antibody diluted in a fresh blocking buffer. Next day membranes were washed three times for 10 min in the wash buffer, incubated with HRP-conjugated secondary antibody (see Supplementary Data 3 for antibody information) for an hour at room temperature (RT), and washed again three times in the wash buffer. SuperSignal West Pico Plus Chemiluminescence Substrate (Pierce) was used for detection and digital images were acquired using an ImageQuant LAS 4000 (GE Healthcare Life Sciences). All uncropped blots are provided in the Source Data files.

## Flow cytometry

The fraction of apoptotic cells was determined by Annexin V-FITC/PI assay. Briefly, cells harvested by trypsinization were resuspended in Annexin-V binding buffer (10 mM HEPES pH 7.4, 140 mM NaCl, 2.5 mM CaCl$_2$). Approximately $2 \times 10^5$ cells were labeled with Annexin-V-FITC (Invitrogen) and PI (10 μg/ml, Millipore-Sigma) for 15 min in the dark before flow cytometric analysis (Accuri C6, Becton Dickinson). Acquired data were analyzed using Accuri C6 software (version 1.0.264.21) and visualized with FlowJo (version 10.4.2).

To analyze mitochondrial membrane potential (ΔΨm) cells were trypsinized and resuspended in cultivation media. An aliquot of approximately $5 \times 10^5$ cells per sample was mixed with Tetramethylrhodamine, Ethyl Ester, Perchlorate (TMRE, Thermo Fisher, 100 nM final concentration) solution, incubated for 10 min in the dark, and analyzed by flow cytometer. Reactive oxygen species levels were measured using 6-chloromethyl-2′,7′-dichlorodihydrofluorescein diacetate, acetyl ester (CM-H$_2$DCFDA). Briefly, trypsinized cells were resuspended in the cultivation media, combined with CM-H$_2$DCFDA solution (10 μM final concentration), and incubated for 15 min in the dark. At least $10^4$ particles per sample were analyzed for fluorescence intensity in the FL1 channel (533/30 nm). Proteasomal activity was analyzed with Me4BodipyFL-Ahx3Leu3VS fluorescent probe (abbreviated as Me4BodipyFL). After the treatment period, cultivation media in both TPC1 and HCT116 cells was replaced with pre-warmed 0.5 μM of Me4BodipyFL in PBS for 1 h. Next, cells were harvested by trypsinization, and fluorescence was measured by flow cytometry. Intracellular levels of Fe$^{2+}$ ions were measured with FerroOrange probe (Dojindo). Briefly, trypsinized cells were resuspended in HBSS buffer, pelleted, resuspended in serum-free DMEM media, and stained with 1 μM FerroOrange dye for 15 min at 37 °C. Gating strategy is shown in Supplementary Fig. 8.

## Q-RT-PCR

After indicated treatments, cells were washed with PBS, total RNA was extracted using Trizol substitute (38% Phenol, saturated, pH 4.3, 0.8 M guanidine thiocyanate, 0.4 M ammonium thiocyanate, 0.1 M sodium acetate, pH 5.0, 5% glycerol), and converted to cDNA with High-Capacity cDNA Reverse Transcription Kit (Thermo Fisher). Next, diluted cDNA was used in quantitative PCR reaction using SYBR Select Master Mix for CFX (Thermo Fisher). Detected mRNA levels were normalized to 18s rRNA values. All primers are listed in Supplementary Data 3.

## RNA-seq library preparation, sequencing, and data analysis

TPC1 and K1 cells were plated at $2 \times 10^4$/cm$^2$ and treated for 24 h as indicated. Following the treatment period, cells were washed with ice-cold PBS and lysed in TRI Reagent (Millipore Sigma). Quality of the extracted RNA was assessed by Agilent Bioanalyzer 2100 using RNA 6000 Pico chips (Agilent). Single-end 150 bp sequencing of the poly-A(+)-enriched RNA was carried out on the Illumina HiSeq 4000 platform by the Genomics Core facility at the University of Colorado Anschutz.

Quality of the sequencing data was analyzed using FASTQC (version 0.11.2, https://www.bioinformatics.babraham.ac.uk/projects/fastqc/) and presence of common sequencing contaminants was assessed by FastQ Screen (v0.4.4) (https://www.bioinformatics.babraham.ac.uk/projects/fastq_screen/). Bases with low quality (Q < 10) were 3′ end trimmed and reads shorter than 30 nt were discarded using the Fastx toolkit (v0.0.13.2). Reads were aligned to a GRCh37/hg19 Human reference using TopHat2 (v2.0.13, --b2-sensitive --keep-fasta-order --no-coverage-search --max-multihits 10 --library-type fr-firststrand)[98] with the UCSC hg19 GTF annotation file provided in the iGenomes UCSC hg19 bundle (https://support.illumina.com/sequencing/sequencing_software/igenome.html). Aligned reads with MAPQ < 10 quality were removed using SAMtools (v0.1.19). Alignments were then sorted by coordinates, and duplicates were identified using Picard (v1.129). Quality assessment of final mapped reads was conducted using RSeQC (v2.6)[99]. Gene-level counts were obtained using HTSeq (v0.6.1)[100] with the following options (--stranded=reverse −minaqual=10 −type=exon −idattr=gene_id --mode=intersection-nonempty) using the iGenomes UCSC hg19 GTF annotation file. Differential gene expression was evaluated using DESeq2 (version 1.6.3)[101] in R (version 3.1.0), using q < 0.05 (FDR < 5%) and fold-change >1.5 (Up) or <1/1.5 (Down) as cutoffs for differentially expressed genes. Genome browser snapshots were generated from bedGraph or tgv files using IGV genome viewer (v2.8.10)[102].

## Chromatin immunoprecipitation, ChIP-seq library preparation, sequencing, and data analysis

Sub-confluent cultures of TPC1 and K1 cells were treated for 24 h with indicated compounds. After the treatment period, cultivation media was replaced with crosslinking solution (1% formaldehyde in PBS) and plates were incubated for 15 min at RT. Next, formaldehyde was quenched with glycine (0.125 mM final) for 5 min and cells were washed twice with ice-cold PBS. Crosslinked cells were lysed in RIPA

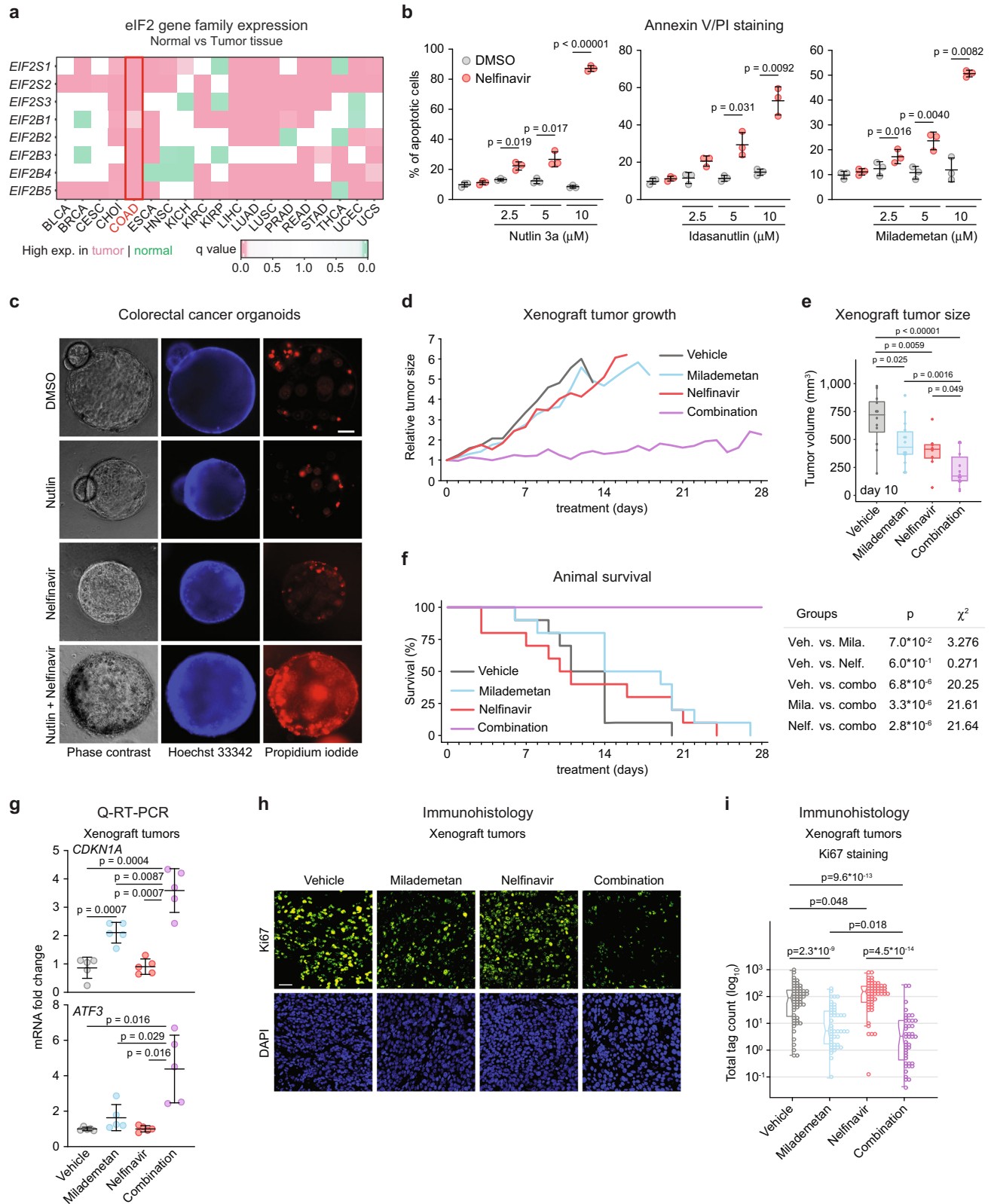

buffer (150 mM NaCl, 50 mM Tris pH 8.5 mM EDTA, 1% IGEPAL 630 (NP-40 substituent), 0.5% sodium deoxycholate, 0.1% SDS and protease/phosphatase inhibitors) and sonicated to generate 200-300 bp fragments of DNA (Qsonica Q800R, 70% amplitude, 30 sec on/30 sec off cycle, 20 cycles for TPC1 lysates and 25 cycles for K1 lysates). Next, samples were centrifuged at 20,000 × g for 20 min at 4°C, protein concentration in collected supernatants was measured using a BCA

Protein Assay Kit and all samples were diluted to final protein concentration of 1 mg/ml. Lysates were pre-cleared with 15 μl of Dynabeads M-280 (sheep anti-mouse IgG, Thermo Fisher Scientific) and immunoprecipitated overnight either with 5 μl of anti-p53 antibody (DO-1, EMD Millipore), or 5 μl of anti-ATF4 antibody (sc-390063X, Santa Cruz Biotechnology), or 50 μl of normal IgG (sc-2025, Santa Cruz Biotechnology) using 30 μl of Dynabeads per reaction. In total, 4

**Fig. 5 | Reduced tumor growth and extended survival by combined inhibition of MDM2 and eIF2α. a** Normalized mRNA expression of eIF2 complex subunits in normal and tumor samples obtained from TCGA and GTEx databases. Statistical significance was calculated using Wilcoxon signed-rank test. BLCA: bladder urothelial carcinoma ($n$ normal = 28, $n$ tumor = 362 independent samples), BRCA: breast invasive carcinoma ($n$ normal = 199, $n$ tumor = 982 independent samples), CESC: cervical squamous cell carcinoma and endocervical adenocarcinoma ($n$ normal = 9, $n$ tumor = 31 independent samples), COAD: colon adenocarcinoma ($n$ normal = 380, $n$ tumor = 285 independent samples), ESCA: esophageal carcinoma ($n$ normal = 278, $n$ tumor = 183 independent samples), HNSC: head and neck squamous cell carcinoma ($n$ normal = 42, $n$ tumor = 460 independent samples), KICH: kidney chromophobe carcinoma ($n$ normal = 57, $n$ tumor = 60 independent samples), KIRC: kidney renal clear cell carcinoma ($n$ normal = 104, $n$ tumor = 475 independent samples), KIRP: kidney renal papillary cell carcinoma ($n$ normal = 61, $n$ tumor = 236 independent samples), LIHC: liver hepatocellular carcinoma ($n$ normal = 163, $n$ tumor = 295 independent samples), LUAD: lung adenocarcinoma ($n$ normal = 372, $n$ tumor = 503 independent samples), LUSC: lung squamous cell carcinoma ($n$ normal = 364, $n$ tumor = 489 independent samples), PRAD: prostate adenocarcinoma ($n$ normal = 154, $n$ tumor = 426 independent samples), READ: rectum adenocarcinoma ($n$ normal = 349, $n$ tumor = 87 independent samples), STAD: stomach adenocarcinoma ($n$ normal = 225, $n$ tumor = 380 independent samples), THCA: thyroid carcinoma ($n$ normal = 371, $n$ tumor = 441 independent samples), UCEC: uterine corpus endometrial carcinoma ($n$ normal = 105, $n$ tumor = 141 independent samples), UCS: uterine carcinosarcoma ($n$ normal = 82, $n$ tumor = 47 independent samples). **b** HCT116 cells were treated with 20 μM nelfinavir and indicated MDM2 inhibitors for 48 h, stained with Annexin V-FITC/PI and analyzed by flow cytometry. Data are represented as mean ± SD. Statistically significant differences ($n$ = 3 independent experiments) were calculated by paired, two-sided $t$ test. **c** PDX-derived line CRC172 colorectal cancer organoids were treated for 48 h with vehicle (0.2% DMSO), 10 μM nutlin-3a, 20 μM nelfinavir, and drug combination. Organoids were live-stained with propidium iodide and Hoechst 33342. Scale bar is 50 μm long. **d** Therapeutic effects of milademetan (200 mg/kg), nelfinavir (200 mg/ml), and drug combination administered by oral gavage once daily 5 days/week in nude mice bearing HCT116 xenograft tumors. Average initial tumor size was $183 \pm 127$ mm$^3$ (see Supplementary Fig. 5b). Ten animals per treatment group were used in the study and datapoints with $n \geq 3$ were plotted. Relative tumor sizes represent average volumes of both tumors in the animal. **e** Comparison of tumor volumes measured at day 10 following the indicated treatments. Tumors from each flank are plotted as individual values. Statistically significant differences ($n$ vehicle = 12, $n$ milademetan = 14, $n$ nelfinavir = 8, $n$ combination = 14) were calculated by unpaired, two-sided $t$ test. **f** Animal survival of nude mice carrying HCT116 xenograft tumors. Animals were sacrificed at the humane endpoint when tumor volume exceeded 1000 mm$^3$. Statistical significance was calculated by log-rank test. **g** Q-RT-PCR of *CDKN1A* and *ATF3* mRNAs in RNA extracted from tumors. Data are represented as mean ± SD. Unpaired, two-sided $t$ test was used to calculate the indicated $p$ value ($n$ = 5 tumor samples from 5 individual animals per group). **h**, **i** Formalin-fixed xenograft tumors were stained with DAPI and Ki67 antibody. Nuclei were identified and scored using InForm software. Scale bar in **h** is 50 μm long. Statistical significance in **i** was calculated using Wilcoxon signed-rank test ($n$ = 55 fluorescence values in 55 randomly subsampled cells from >19,900 per group. Histology samples were obtained from 3 animals per group). Box plots center lines in **e**, **i** represent median values, box boundaries outline the 25th and 75th percentile. Whiskers depict the smallest or largest values within 1.5 times of the interquartile range. See also Supplementary Fig. 7. Source data are provided as a Source data file.

(TPC1) or 5 (K1) lysate aliquots per sample were used in immunoprecipitation reactions for ChIP-seq sample preparation. Next day beads were washed (5 min each washing step) twice with RIPA, four times with IP wash buffer (500 mM LiCl, 100 mM Tris pH 8.5, 1% IGEPAL, 1% sodium deoxycholate), again twice with RIPA and twice briefly with TE (10 mM Tris pH 8, 1 mM EDTA). Washed beads were resuspended in 100 μl of TE and 200 μl of elution buffer (70 mM Tris pH 8, 1 mM EDTA and 1.5% SDS) and incubated at 65 °C for 10 min. After adding NaCl to final concentration of 200 mM, eluted immunocomplexes were incubated at 65 °C for 5 h to reverse formaldehyde crosslinks. Remaining protein was digested by proteinase K (20 μg/sample, 45 °C for 30 min). DNA was recovered by one phenol/chloroform and one chloroform extraction followed by ethanol precipitation and resuspension in 50 μl of TE. Input DNA was extracted from reverse cross-linked lysates using the same extraction protocol as for sample DNA.

ATF4 ChIP samples were analyzed by Q-PCR with primers listed in Supplementary Data 3 designed based on ATF4 occupancy in K562 cell line (ENCODE dataset ENCSR044UJJ).

Precipitated DNA fragments for the p53 ChIP-seq analysis were size-selected (80-600 bp) using agarose gel electrophoresis (2% gel, BluePippin) and barcoded with the NEBNext Ultra II DNA sequencing library preparation kit, according to the manufacturer's instructions (New England Biolabs). Next, libraries were size-selected (200–600 bp, BluePippin) and analyzed on Bioanalyzer High Sensitivity DNA chips (Agilent) to confirm 200–400 bp fragment size range. Single-end 150 bp sequencing of pooled barcoded libraries was carried out on the Illumina HiSeq 4000 platform by the Genomics Core facility at the University of Colorado Anschutz.

ChIP-seq data quality was assessed using FASTQC (v0.11.5) and FastQ Screen (v0.11.0). Trimming and filtering of low-quality reads was performed using FASTQ-MCF from EAUtils (v1.05). Alignment to the human reference genome (GRCh37/hg19) was carried out using Bowtie2 (v2.2.9)[103] in sensitive end-to-end mode with a GRCh37/hg19 index, and alignments were sorted and filtered for mapping quality (MAPQ > 10) using Samtools (v1.5)[104]. Alignments were then coordinate sorted, and duplicates were marked using Picard (v2.9.4). Quality assessment of final mapped reads was conducted using RSeQC

(v2.6.4)[99]. BigWig files for visualization of p53 occupancy were generated with deepTools[105] (version 2.2.2, settings --binSize=1 –extendReads FRAGMENT_LENGTH --minMappingQuality 10 –normalizeUsingRPKM). Read density was displayed at 1 bp resolution as reads per million of mapped reads per 1 kb (RPM/kb).

### Recombinant ATF4 expression
Total RNA was extracted from TPC1 cell line by Trizol substitute, treated with RQ1 DNaseI (Promega, Fisher Scientific), and used as a template in reverse transcription reaction (SuperScript IV, Invitrogen, Thermo Fisher). Next, ATF4 cDNA was amplified by PCR (Phusion High Fidelity DNA polymerase, Fisher Scientific, see Supplementary Data 3 for primer sequences) and cloned into the pJET1.2 blunt end cloning vector. SalI and NotI restriction sites were used for transferring the insert to pENTR4 and pLenti CMV Tet-on vector.

### MTT assays
Metabolic activity assay based on converting tetrazolium salt to formazan[106] was carried out in 96 well plates. Cells plated at density $2 \times 10^4$/cm$^2$ were cultivated o/n and exposed to tested inhibitors for 72 h in triplicates. Solution of 2.5 mg/ml MTT (3-(4,5-dimethylthiazol-2-yl) −2,5-diphenyltetrazolium Bromide) was prepared in PBS and added to cultivation media at final concentration 0.25 mg/ml. After 1 hour incubation at 37 °C was the mixture replaced with 100 μl of lysis buffer. Next, plates were placed on an orbital shaker. Following a complete dissolution of formazan crystals absorbance was measured at 570 nm. Absorbance values from using MTT assays were analyzed with SynergyFinder software[107]. The degree of synergy was calculated using highest single agent (HSA) reference model. Synergy scores of specific concentrations (δ-score) was plotted to show synergy distribution. The higher the synergy score, the stronger the synergy at those concentrations.

### Polysome profiling
TPC1 and HCT116 were plated at $6 \times 10^4$/cm$^2$ on 143 cm$^2$ dishes, cultivated o/n, and treated as indicated in the various figure legends. Ten minutes before the harvest cultivation media was supplemented with cycloheximide (CHX) at final concentration of 100 μg/ml. Next, cells

were washed twice with ice-cold PBS with 100 μg/ml CHX and lysed in polysome preparation lysis buffer (20 mM HEPES pH 7.4, 15 mM MgCl₂, 200 mM NaCl, 1% Triton X-100, 100 μg/ml CHX, 2 mM DTT, and 100 U SuperaseIN). Lysates were cleared of debris by centrifugation at $20,000 \times g$, 4 °C for 10 min. Total nucleic acid content in lysates was measured by absorbance at 260 nm and used for sample concentration normalization. Next, 500 μl of the lysate was loaded on 10–60% sucrose gradients in SW41 tubes in lysis buffer lacking Triton X-100. These gradients were prepared using a BioComp system and chilled to 4 °C before use. Samples were ultracentrifuged at $160,000 \times g$ for 3 h and 10 min, at 4 °C, then samples were fractionated using a BioComp system, monitoring absorbance at 260 nm while collecting fractions of approximately 0.4 ml each.

## Regulatory heme analysis

Intracellular regulatory (free) heme was measured with established protocols[108] as follows: after the treatment period, cells were washed twice with PBS, lysed in 0.1% Triton X-100 in PBS supplemented with protease inhibitors, scrapped, and transferred to Eppendorf tubes, briefly sonicated (2.5 W, 5 seconds), and centrifuged at $18,000 \times g$, 4 °C for 10 min. Next, 10 μl (TPC1, HCT116) or 30 μl (SJSA) of the lysate was combined with 100 μl of 5 μM apoHRP, 100 μl of 1.25 μM TMB, and 50 μl of 10 mM $H_2O_2$ in PBS. Following 5 min incubation absorbance was measured at 352 mm. Protein concentration in lysate aliquots were analyzed by BCA kit and resulting values were used to correct heme level readouts for differences in sample densities.

## Immunohistochemistry

Tumors fixed in 4% formaldehyde were transferred to 70% ethanol and embedded in paraffin at the Pathology Shared Resource–Research Histology, CU-AMC. Three tumors from each experimental group were sectioned, stained with DAPI and Ki67 primary antibody using Akoya Opal technology (Akoya Biosciences), and scanned at six representative regions with Vectra 3.0 (Akoya Biosciences) at the Human Immune Monitoring Shared Resource (CU-AMC). Next, InForm image analysis software (Akoya Biosciences) was used for automated identification of nuclei based on the DAPI signal. Ki67 nuclear fluorescence was outputted for all nuclei. Resulting libraries were downsampled based on power analysis calculation[109] using the formula $n = (Z\sigma/E)^2$ where n is the sample size required to ensure that the margin of error ($E$, 95%) does not exceed the value specified as 25% of the vehicle-treated nuclei signal, $Z$ is the value from the table of probabilities of the standard normal distribution for the desired confidence level, and $\sigma$ is the standard deviation of the outcome of interest. For antibody information see Supplementary Data 3.

## Statistics and reproducibility

Data are presented either as one-dimensional scatter plots showing mean ± standard deviation (SD) or standard box-and-whisker plots. Briefly, center horizontal line denotes median value, boxes above and below are outlined by the upper and lower data quartile, respectively. Notches represent confidence intervals around median values. Whiskers show data range from interquartile range to maximum and minimum values, excluding outliers. Data were graphed in R (version 3.1.0) using *ggplot2* library.

For comparison between two groups, datasets were analyzed by either two-tailed Student's $t$ test, hypergeometric test, Fisher's exact test, Wilcoxon signed-rank test, or log-rank test as indicated. All measurements were taken from multiple independent biological replicates as indicated in each figure legend.

## Inclusion and ethics statement

We ensured sex balance in the selection of non-human subjects. One or more authors of this study self-identifies as a member of a minority underrepresented in science.

## Reporting summary

Further information on research design is available in the Nature Portfolio Reporting Summary linked to this article.

## Data availability

The data that support this study are available from the corresponding authors upon request. RNA-seq and ChIP-seq data generated in this study have been deposited to the Gene Expression Omnibus (GEO) database and are available under the accession number GSE191150. This paper analyzed data from the Genotype-Expression Project (GTEx) [https://gtexportal.org/home/datasets], The Cancer Genome Atlas Project (TCGA) [https://www.cancer.gov/about-nci/organization/ccg/research/structural-genomics/tcga/using-tcga], as well as publicly available data for ATF4 chromatin binding ENCSR044UJJ and global run on-deep sequencing (GRO-seq) data and matching RNAseq data under conditions of p53 stimulation GSE86222. Microscopy images are shared at FigShare portal [https://figshare.com/articles/media/mouse_tumors_and_organoids/21545292]. Any additional information required to re-analyze data reported in this paper will be provided by the corresponding authors upon request. Source data are provided with this paper.

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

## Acknowledgements

This work was supported primarily by NIH grant 5R01CA117907 (J.M.E).
Additional support was provided by Cancer League of Colorado grant
AWD-203708-AZ (Z.A.), American Cancer Society Institutional Research
grant 16-184-56 (Z.A.), and NIH grant R35GM118070 (J.S.K.). We thank
Maria Szwarc, Santosh Khanal, Molishree Joshi, and Emily Adams for
technical assistance, and Dr. Peter Dempsey for providing the colorectal
cancer organoid used in this study. This project employed shared
resources supported by the University of Colorado Cancer Center via
grant P30CA046934.

## Author contributions

J.M.E. and Z.A. conceived the experiments and wrote the manuscript.
Z.A. performed most of the experiments and analyzed data. J.S.K. per-
formed polysome fractionations. K.D.S. assisted with in vivo experi-
ments. All authors reviewed and edited the manuscript.

## Competing interests

J.M.E. has provided consulting services for Elli Lily and Co. and Gilead
Sciences Inc. and serves in the advisory board of Perha Pharmaceuticals.
J.M.E. and Z.A. have applied as co-inventors in a provisional patent for
employing dual activation of the p53 and ISR networks in cancer therapy
(U.S. Provisional Patent Application 63/356,432). The other authors
declare no competing interests.

## Additional information

**Supplementary information** The online version contains
supplementary material available at

Zdenek Andrysik or Joaquin M. Espinosa.

**Peer review information** *Nature Communications* thanks Martin Fischer
and the other, anonymous, reviewer(s) for their contribution to the peer
review of this work. Peer reviewer reports are available

