## [Peer Review File · Nature Communications]

REVIEWER COMMENTS

Reviewer #1 (Remarks to the Author):

Andrýsik et al. investigate the mechanisms that underlie p53-dependent cell death induced by dual inhibition of the negative p53 regulators MDM2 and PPM1D. The authors corroborate earlier findings whereby PPM1D inhibition increased the sensitivity of MDM2 inhibitors to induce p53-dependent apoptosis. Importantly, the authors identify eIF2a, a translation initiation factor, to be inhibited in response to MDM2/PPM1D dual inhibition. When PPM1D is inhibited in addition to MDM2, the eIF2a upstream kinase HRI becomes activated, presumably following heme reduction by elevated HMOX1 levels, leading to decreased overall translation but specifically increased translation of the transcription factor ATF4. As a downstream event, ATF4 activates its transcriptional program in which it cooperates in part with p53 to boost p53 target gene expression. Perhaps most importantly, solving this molecular mechanism led the authors to identify the eIF2a modulatory drugs Nelfinavir and Sal003 as promising candidates for combination therapy with MDM2 inhibitors.

Overall, the study solves an important and intriguing molecular mechanism that underlies combined MDM2 and PPM1D inhibition with potential relevance for clinical applications. The experiments and the manuscript are of high quality and, in general, the study merits publication in Nature Communications. However, I felt that a few important experiments and controls are missing, before I can recommend its publication.

Major:

- Given that the authors identified a prominent role of the transcription factor ATF4, I felt that an assessment of ATF4 binding (ChIP-seq) was missing. ATF4 ChIP-seq data is also publicly available (Cohen et al. 2015 eLife and ENCODE). Integrating ATF4 ChIP-seq data would provide a more detailed picture of what part of the transcriptional program induced by dual inhibition was more likely to be affected by ATF4 itself. ATF4 ChIP-qPCR at some individual targets investigated by the authors could further validate a potential direct role of ATF4.
- The authors introduce and discuss the potential importance of the ATF4-ATF3 axis, but they did not address whether ATF3 affected the transcriptional program. To this point, at least Figure 2h and S2f should include shATF3. RNA-seq of Nutlin+GSK treated cells upon ATF4 and ATF3 depletion would be ideal, but not strictly necessary.
- I felt that the authors rather jumped from elevated HRI levels (Figure 3e) to HRI-dependent ATF4 induction (Figure 3f). Before examining ATF4 levels, I felt that the authors should check p-eIF2a levels. After all, their model states that HRI activates ATF4 through increasing p-eIF2a. Moreover, I felt that a knockdown of the other three kinases was missing as a control in Figure 3f and should be included also for an additional sub-Figure that checks p-eIF2a levels. Particularly of PERK, which actually appears to be increased too (Figure S3a) – at least to my eyes, the authors state it is not.
- I felt that more direct evidence was missing for the HMOX1-heme-HRI signaling axis suggested by the authors. Is heme depletion and HRI-eIF2a-ATF4 activation upon MDM2/PPM1D inhibition impaired when HMOX1 is depleted?
- Given that Nutlin is unlikely to be used in the clinic for its known issues in bioavailability, the authors should corroborate some of the effects of dual MDM2/PPM1D and MDM2/eIF2a inhibition using next generation MDM2 inhibitors. Such as by including RG7112 and RG7388 in Figures S1c and S4a-c.

Minor:

- Additional known limitations of MDM2 inhibitors should be mentioned in the introduction and/or discussion. E.g. low bioavailability of Nutlin and toxicity of next generation MDM2 inhibitors (Ray-Coquard et al. 2012 Lancet Oncol).

- The authors note that PPM1D mutations are mutually exclusive to TP53 mutations in thyroid cancer. It would be helpful to the reader if the authors indicated the type of mutations (i.e. activating mutations).
- Figure 1f would be more informative if it contained a set of established common p53 targets, such as the 103 core genes identified in the authors' 2017 study published in Genome Research, instead of largely recapitulating the examples already shown in 1e.
- At the end of results section two, the authors conclude that "inhibition of MDM2 and PPM1D induces the AFT4-ATF3 axis", although the authors did not test for an ATF4-ATF3 axis but only for the individual proteins.
- The authors should explain already in the results section what eIF2a phosphorylation they investigated (currently it is to be found in discussion). Along those lines, I couldn't find the antibodies used by the authors in the methods section.
- Figure 4g: The visualization by SynergyFinder is not intuitive. It is unclear why the synergy (green/red shading) appears to not depend on the concentration or even the presence of Nutlin (the green/red edge is seen at the same concentration of Nelfinavir/Sal003 regardless of Nutlin concentration, including 0 Nutlin).
- I felt that Aziz et al 2011 Oncogene should be referenced when discussing MDM2 inhibitor-induced p53 mutations.
- I felt that a discussion on Sal003 was missing.

Reviewer #2 (Remarks to the Author):

In this manuscript Andrysiak et al. describe a synergy between p53 activating drugs and the integrated stress response to achieve anti-cancer effects both in vitro and in vivo. Reactivation of the tumor suppressor p53 is a holy grail in cancer research. In tumors, p53 can be mutated leading to a loss of function (and perhaps gain of function), or suppressed via proteins modulating its stability. Consequently, there are molecules reactivating mutant p53 as well as reversing p53 degradation. In this study the Espinosa lab focuses on the latter category where nutlins are considered state of the art compounds reactivating p53 in tumors where p53 is degraded due to high MDM2 expression. Although such drugs are excellent in reactivating p53 they only induce cell cycle arrest and dormancy which leads to tumor cell escape. Accordingly, approaches to convert the cell-cycle inhibitory phenotype to a cell-death inducing phenotype is of utmost importance in the field. p53 is activated following a number of stresses and its thought to allow the cell to recover from the stress or target the cell for death in absence of recovery. Intriguingly, the integrated stress response has a similar role but responds to a distinct set of stresses. These studies suggest that inducing multiple stress pathways can synergize to kill cancer cells, without affecting normal cell function (as judged by mouse weight). Although it appears that the means to activate the integrated stress response is not essential for this activity the authors also describe PPM1D-dependent activation of the integrated stress response depending on HRI which in turn phosphorylates eIF2a in response to multiple signals. The authors further suggest that heme depletion downstream of PPM1D-inhibition underlies the observed synergy between PPM1D-inhibition and p53 reactivation. Overall, the studies are of high quality and the results will be of high interest to those working on cancer, mRNA translation and cell biology. I have a few suggestions which to me would strengthen the manuscript further.

Major concerns:

1. Although I find the informatics in figure 1 convincing the comparisons done in figure 1d-e are insufficient. To identify genes that are targets by the combination but not nutlin alone, statistical analysis should be performed to compare these conditions. Using Venn diagrams to identify sets differing between treatments is very threshold sensitive (i.e. just blow a threshold in one condition relative control and just above in the second condition relative to

control would be seen as distinct between the conditions whereas if the direct comparison had been made (i.e. between the two treatments) this would not have been concluded. Furthermore, only relying on fold-changes seems insufficient.

2. There is no data showing that the ATF4 mRNA shifts towards increased association with polysomes following treatments (i.e. in relation to figure 3c). This would normally be compared to an mRNA which does not shift.

3. As the integrated stress response involves massive modulation of mRNA translation it seems important to quantify changes in translation using RNA sequencing across the treatments using e.g. polysome-profiling. Does p53 activation potentiate the integrated stress-response?

4. Western blotting suggest that PERK, PKR and GCN2 do not undergo increased phosphorylation of eIF2alpha. This should be further substantiated by using available inhibitors towards these kinases.

5. There are transcriptional signatures of from prolonged activation of the integrated stress response (Han et al Nature Cell Biol 2013). How are these transcriptional programs regulated under the treatments?

6. Does the observed apoptosis depend on CHOP, a downstream target of the integrated stress response, or apoptosis inducers downstream of p53? This would address the broader question of whether ISR or p53-dependent cell death which is activated.

7. Is it possible to rescue the heme reduction to show that this is truly underlying HRI activation? For example, by example silencing expression of HMOX1?

8. The use of the MTT assay could be problematic as it measures metabolic activity which may also be affected by the treatments. Therefore, MTT assay may not reflect cell viability.

9. There is earlier literature suggesting that apoptosis following treatment with p53 reactivation drugs requires activation of the integrated stress response (Johannes Ristau, Cell Death and Disease 2019 & Jun Yang, Mol Cell Biol 2009). This could be included in the discussion as it may indicate that this is a general feature.

Minor concerns:

1. Figure 1i. The format "X versus Y" would normally mean that fold changes would be calculated via X/Y (in log scale). Here it seems to be the other way around.

2. The authors occasionally use "protein translation" but proteins are not translated. The correct term is "mRNA translation" or "protein synthesis".

Reviewer #3 (Remarks to the Author):

In this manuscript, the authors found that dual inhibition of MDM2 and eIF2A induces apoptosis in vitro and decreases tumor growth in mouse xenograft models.

The study is well-executed and thoroughly elucidates the combined roles of the p53 and eIF2A pathways in apoptosis. However, the connection between PPM1D inhibition with GSK2830371 on ATF4 protein accumulation through eIF2A phosphorylation could be further developed. When MDM2 is inhibited and ATF4 is induced through eIF2A phosphorylation with GSK2830371 or nelfinavir, there is clear evidence for increased apoptosis. This is suggesting a greater role of ATF4 induction in this phenotype as opposed to PPM1D inhibition. The subject of the manuscript could be adjusted accordingly to acknowledge this. Additionally, the following suggestions will improve the manuscript before acceptance in Nat Comm.

Major comments:

1. Because PPM1D has been previously shown to dephosphorylate p53 at Ser15, perform a Western blot for Phospho-p53 (Ser15) after PPM1D inhibition with GSK2830371 as in Fig.

1A.

2. To support the claim that ATF4 can undergo selective translation, measure ATF4 mRNA association with collected polysome fractions in Fig. 3C and 4C.

3. There are inconsistencies in the treatment times between similar experiments, making it difficult to make comparisons. Address the significant decrease in ATF4 mRNA expression after Nutlin treatment for 24 hours (Fig. S2A-B). Provide RT-qPCR analysis for ATF4 mRNA expression after 6 hours of Nutlin treatment as in Fig. S2A and a Western blot for ATF4 as in Fig. S2B after 24 hours of Nutlin treatment.

4. Fig 2b: A panel with ATF4 ChIP on ATF3 gene is required to show how ATF4 regulates ATF3 expression.

5. Fig 2i: Although there is a trend for upregulation of p53 target genes after ATF4 overexpression, it does not seem to be significant based on the error bars. They should either reanalyze their results or increase their number of repeats to decrease the error bars

6. Fig S3b: How does EIF2AK1 mRNA levels go down after Nutlin+GSK administration while the protein levels go up (Figure 3b)? The authors should explain this unexpected result

7. Fig S3f-g: The authors claim that the mitochondrial membrane potential is not changing but the graphs show the opposite. Are the changes not statistically significant?

8. Fig 4e: What happens to ATF3 protein levels after Nutlin and Nelfinavir treatment? Is it the same as GSK+Nutlin?

Minor comments:

1. Some references to figure panels are missing from the text.

2. Statistics for Fig. S3D and S3F are missing.

Response to Referees Letter – NCOMMS-2200886

Point by Point response to Reviewers' Comments.

Reviewer 1 (Remarks to the Author):

Andrysyk et al. investigate the mechanisms that underlie p53-dependent cell death induced by dual inhibition of the negative p53 regulators MDM2 and PPM1D. The authors corroborate earlier findings whereby PPM1D inhibition increased the sensitivity of MDM2 inhibitors to induce p53-dependent apoptosis. Importantly, the authors identify eIF2a, a translation initiation factor, to be inhibited in response to MDM2/PPM1D dual inhibition. When PPM1D is inhibited in addition to MDM2, the eIF2a upstream kinase HRI becomes activated, presumably following heme reduction by elevated HMOX1 levels, leading to decreased overall translation but specifically increased translation of the transcription factor ATF4. As a downstream event, ATF4 activates its transcriptional program in which it cooperates in part with p53 to boost p53 target gene expression. Perhaps most importantly, solving this molecular mechanism led the authors to identify the eIF2a modulatory drugs Nelfinavir and Sal003 as promising candidates for combination therapy with MDM2 inhibitors.

Overall, the study solves an important and intriguing molecular mechanism that underlies combined MDM2 and PPM1D inhibition with potential relevance for clinical applications. The experiments and the manuscript are of high quality and, in general, the study merits publication in Nature Communications. However, I felt that a few important experiments and controls are missing, before I can recommend its publication.

Major:

- Given that the authors identified a prominent role of the transcription factor ATF4, I felt that an assessment of ATF4 binding (ChIP-seq) was missing. ATF4 ChIP-seq data is also publicly available (Cohen et al. 2015 eLife and ENCODE). Integrating ATF4 ChIP-seq data would provide a more detailed picture of what part of the transcriptional program induced by dual inhibition was more likely to be affected by ATF4 itself. ATF4 ChIP-qPCR at some individual targets investigated by the authors could further validate a potential direct role of ATF4.

Response: Thanks for this comment, which promoted new analyses, experiments, and results in the revised manuscript. Analysis of ChIP-seq data for ATF4 available through the ENCODE project revealed a clear pattern of ATF4 binding preferentially to genes that are more strongly induced upon combined inhibition of MDM2 and PPM1D. Indeed, ATF4 binds to ~38% and ~60% of the genes more strongly induced by the combinatorial treatment in K1 and TPC1 cell lines, respectively, but only to ~8% and 18% of those who are not further induced by PPM1D inhibition. This is now shown in new **Figure S3d**. Additionally, we completed ChIP-qPCR experiments for ATF4 in our experimental paradigm as suggested by the Reviewers and confirmed that indeed ATF4 binds to predicted ATF4 response elements (ATF4 REs) at p53 target genes that are over-induced upon dual inhibition of p53 repressors, such as ATF3, BBC3/PUMA, and GDF1. These results are shown in new **Figure 2j**.

- The authors introduce and discuss the potential importance of the ATF4-ATF3 axis, but they did not address whether ATF3 affected the transcriptional program. To this point, at least Figure 2h and S2f should include shATF3. RNA-seq of Nutlin+GSK treated cells upon ATF4 and ATF3 depletion would be ideal, but not strictly necessary.

Response: Thanks for this comment, which elicited new experiments and results. Following Reviewer's guidance, we knocked down ATF3 expression with two different shRNAs and monitored the impact of ATF3 depletion on the p53 transcriptional program. Expectedly, ATF3 depletion reduced induction of several p53 target genes that we previously showed required ATF4 for full expression. These results are shown in new **Figure S3c**.

- I felt that the authors rather jumped from elevated HRI levels (Figure 3e) to HRI-dependent ATF4 induction (Figure 3f). Before examining ATF4 levels, I felt that the authors should check p-eIF2a levels. After all, their model states that HRI activates ATF4 through increasing p-eIF2a. Moreover, I felt that a knockdown of the other three kinases was missing as a control in Figure 3f and should be included also for an additional sub-Figure that checks p-eIF2a levels. Particularly of PERK, which actually appears to be increased too (Figure S3a) – at least to my eyes, the authors state it is not.

Response: Thanks for these comments, which inspired new experiments and results. First, we now demonstrate that knockdown of HRI with two independent shRNAs produces a significant reduction in both p-eIF2a and ATF4 levels. This is now shown in new **Figure 3f**. Second, knockdown of either PERK, PKR or GCN2 with two independent shRNAs did not impair apoptosis upon dual inhibition of MDM2 and PPM1D. This is now shown in new **Figure S4f-g**. All these new results are referenced in the revised text.

- I felt that more direct evidence was missing for the HMOX1-heme-HRI signaling axis suggested by the authors. Is heme depletion and HRI-eIF2a-ATF4 activation upon MDM2/PPM1D inhibition impaired when HMOX1 is depleted?

Response: Thanks for this comment. In the revised manuscript we show new data demonstrating that HMOX1 knockdown leads to accumulation of regulatory heme leading to levels well above those observed upon dual inhibition of MDM2 and PPM1D, concomitant with loss of the apoptotic phenotype (new **Figure 3j-k**).

- Given that Nutlin is unlikely to be used in the clinic for its known issues in bioavailability, the authors should corroborate some of the effects of dual MDM2/PPM1D and MDM2/eIF2a inhibition using next generation MDM2 inhibitors. Such as by including RG7112 and RG7388 in Figures S1c and S4a-c.

Response: We agree with this comment. In the revised manuscript we demonstrate the synergistic effects of dual inhibition of MDM2 and PPM1D are conserved when using more clinically relevant MDM2 inhibitors such as RG7388 (idasanutlin) and milademetan (currently being tested in Phase III clinical trials). This is now shown in new **Figure S1d**. In agreement with the Reviewer's comment, we used idasanutlin and milademetan in a side-by-side comparison with nutlin in **Figure 5b**, leading to the use of milademetan (not nutlin) in the animal experiments in **Figure 5d-i**.

Minor:

- Additional known limitations of MDM2 inhibitors should be mentioned in the introduction and/or discussion. E.g. low bioavailability of Nutlin and toxicity of next generation MDM2 inhibitors (Ray-Coquard et al. 2012 Lancet Oncol).

Response: We agree with this comment and have revised the Introduction accordingly, including citing the paper referenced by the Reviewer.

- The authors note that PPM1D mutations are mutually exclusive to TP53 mutations in thyroid cancer. It would be helpful to the reader if the authors indicated the type of mutations (i.e. activating mutations).

Response: We agree with this comment and have revised the text accordingly. Gain-of-function mutations in *PPM1D* have now been observed in several cancer types, often with well documented mutual exclusivity with p53 mutations. These mutations most frequently lead to a truncated version of the protein lacking its auto-inhibitory C-terminus domain. These reports are cited in the revised Introduction¹⁻⁵.

- Figure 1f would be more informative if it contained a set of established common p53 targets, such as

the 103 core genes identified in the authors' 2017 study published in Genome Research, instead of largely recapitulating the examples already shown in 1e.

Response: We really liked this comment, which led us to highlight the 103 core direct p53 target genes in **Figure 1e-f**. This also inspired us to create a new panel in this Figure to show how this gene set behaves upon single versus dual inhibition of MDM2 and PPM1D. This analysis revealed that on average this gene set is more induced upon dual inhibition, however, there are clear gene-specific effects, with some genes being more sensitive to PPM1D inhibition than others. In fact, ATF3 was the most 'super-induced' gene upon dual inhibition in TPC1, which further supports the notion of the ATF4-ATF3 pathway being involved in the synergistic action of the two inhibitors. Interestingly, PPM1D was not over-induced by the combinatorial treatment in either cell type.

- At the end of results section two, the authors conclude that "inhibition of MDM2 and PPM1D induces the AFT4-ATF3 axis", although the authors did not test for an ATF4-ATF3 axis but only for the individual proteins.

Response: We agree with this comment and have modified the text accordingly.

- The authors should explain already in the results section what eIF2a phosphorylation they investigated (currently it is to be found in discussion). Along those lines, I couldn't find the antibodies used by the authors in the methods section.

Response: We agree with this comment and apologize for this oversight. We had created a table with all antibody information that was accidentally omitted from the submission. We have revised the text to address this oversight. We clarify that it is indeed serine 51 phosphorylation in the Results and Figure legends. We also provide information on all antibodies used for western blot, flow cytometry and immunocytochemistry in **Supplementary File 3**.

- Figure 4g: The visualization by SynergyFinder is not intuitive. It is unclear why the synergy (green/red shading) appears to not depend on the concentration or even the presence of Nutlin (the green/red edge is seen at the same concentration of Nelfinavir/Sal003 regardless of Nutlin concentration, including 0 Nutlin).

Response: In response to this comment, we have improved the presentation of the Figure (moved all scale labels for the synergy scores outside the graphs for improved visualization) and added additional text to the Methods for greater clarity. The perception that the synergy is also observed a 'Nutlin 0' is not quite correct and driven by the visual effect of the 'topographical map' of the synergy scores. In other words, color is truly 'white' (synergy score 0) along the two axis (i.e., when one of the drugs is absent). We share here with the Reviewer the original visualization of how the synergy plots are created by SynergyFinder as described in Ianevski et al 2020 ⁶:

- I felt that Aziz et al 2011 Oncogene should be referenced when discussing MDM2 inhibitor-induced p53 mutations.

Response: We agree with this comment and have added this citation to the appropriate paragraph in the Introduction.

- I felt that a discussion on Sal003 was missing.

Response: We agree with this comment and have enhanced the description of this compound in the revised Discussion.

Reviewer 2 (Remarks to the Author):

In this manuscript Andrysyk et al. describe a synergy between p53 activating drugs and the integrated stress response to achieve anti-cancer effects both in vitro and in vivo.

Reactivation of the tumor suppressor p53 is a holy grail in cancer research. In tumors, p53 can be mutated leading to a loss of function (and perhaps gain of function), or suppressed via proteins modulating its stability. Consequently, there are molecules reactivating mutant p53 as well as reversing p53 degradation. In this study the Espinosa lab focuses on the latter category where nutlins are considered state of the art compounds reactivating p53 in tumors where p53 is degraded due to high MDM2 expression. Although such drugs are excellent in reactivating p53 they only induce cell cycle arrest and dormancy which leads to tumor cell escape. Accordingly, approaches to convert the cell-cycle inhibitory phenotype to a cell-death inducing phenotype is of utmost importance in the field. P53 is activated following a number of stresses and its thought to allow the cell to recover from the stress or target the cell for death in absence of recovery. Intriguingly, the integrated stress response has a similar role but responds to a distinct set of stresses. These studies suggest that inducing multiple stress pathways can synergize to kill cancer cells, without affecting normal cell function (as judged by mouse weight). Although it appears that the means to activate the integrated stress response is not essential for this activity the authors also describe PPM1D-dependent activation of the integrated stress response depending HRI which in turn phosphorylates eIF2a in response to multiple signals. The authors further suggest that heme depletion downstream of PPM1D-inhibition underlies the observed synergy between PPM1D-inhibition and p53 reactivation. Overall, the studies are of high quality and the results will be of high interest to those working on cancer, mRNA translation and cell biology. I have a few suggestions which to me would strengthen the manuscript further.

Major concerns:

1. Although I find the informatics in figure 1 convincing the comparisons done in figure 1d-e are insufficient. To identify genes that are targets by the combination but not nutlin alone, statistical analysis should be performed to compare these conditions. Using Venn diagrams to identify sets differing between treatments is very threshold sensitive (i.e. just above a threshold in one condition relative to control and just above in the second condition relative to control would be seen as distinct between the conditions whereas if the direct comparison had been made (i.e. between the two treatments) this would not have been concluded. Furthermore, only relying on fold-changes seems insufficient.

Response: We agree with this comment, and the statistical analysis indicated by the Reviewer was indeed presented in the original manuscript, but perhaps not highlighted well enough. The comparison done in the Venn diagrams in **Figure 1d** does not simply rely on fold changes. Each set of genes was defined by **statistically significant induction (FDR 5%)** as defined by DESeq2 and a fold induction >1.5 fold. The statistical analysis of Nutlin versus Nutlin + GSK inhibitor was completed and presented in **Figure S1d (S1e in the updated manuscript)** and **Supplemental File 1** (tabs G and H). In fact, the genes identified through a statistical analysis of Nutlin versus Nutlin + GSK are the ones used in **Figure**

1i to identify ATF4 as the predicted upstream regulator of this gene set. In sum, we are in full agreement with the Reviewer and have revised the text to make clear how this analysis was done.

2. There is no data showing that the ATF4 mRNA shifts towards increased association with polysomes following treatments (i.e. in relation to figure 3c). This would normally be compared to an mRNA which does not shift.

Response: Thanks for this comment, which prompted us to monitor the distribution of ATF4 mRNA in polysomal fractions. Expectedly, upon dual inhibition of p53 repressors, ATF4 mRNA shifts toward heavier polysomal fractions, suggesting an increased translational rate as previously reported during conditions of activation of the ISR. In contrast, the control mRNA GAPDH shifts toward lighter polysomes, consistent with global translational repression. These results are now shown in **Figure S4a**.

3. As the integrated stress response involves massive modulation of mRNA translation it seems important to quantify changes in translation using RNA sequencing across the treatments using e.g. polysome-profiling. Does p53 activation potentiate the integrated stress-response?

Response: We agree with the Reviewer that evaluating global changes in the 'translatome' in the various conditions described in our study would be a worthy pursuit. Our team and others have already investigated changes in the 'translatome' downstream of p53 activation with Nutlin⁷⁻⁹ or neocarzinostatin¹⁰. These studies have identified many mRNAs whose translation rate is affected, either positively or negatively, downstream of p53 activation, including p53 target genes.

Nevertheless, inspired by the Reviewer's comment, we completed an analysis of the key pro-apoptotic p53 target gene BBC3/PUMA. We reasoned that PUMA, which is required for p53-dependent apoptosis in many settings^{11,12}, would bypass the translational repression associated with the ISR and undergo increased translation during conditions of ISR activation. Indeed, Q-RT-PCR analysis of the PUMA mRNA in polysomal fractions demonstrated increased presence of the PUMA mRNA in the 'heavy' fractions (i.e., more strongly translated) along with increased levels of total PUMA protein. These interesting results are shared now in new **Figure 4h-i**. See below response to point #5 about potentiation of the ISR under the various experimental conditions.

4. Western blotting suggest that PERK, PKR and GCN2 do not underly increased phosphorylation of eIF2alpha. This should be further substantiated by using available inhibitors towards these kinases.

Response: Thanks for this comment, which we have addressed with knockdowns for each of these kinases with two independent shRNAs. Unlike knockdown of HRI, which impairs eIF2 phosphorylation, ATF4 accumulation and apoptosis, knockdown of PERK, PKR and GCN2 has no effect on cell viability upon dual inhibition of the two p53 repressors. These new data are shown in new **Figure S4f-g**.

5. There are transcriptional signatures of from prolonged activation of the integrated stress response (Han et al Nature Cell Biol 2013). How are these transcriptional programs regulated under the treatments?

Response: We appreciate this comment (also related to the comment below about CHOP). The paper by Han et al 2013 describes the identification of genes regulated upon induction of ER stress or forced overexpression of ATF4 and CHOP, the two key mediators of transcriptional changes upon ISR activation, in mouse embryo fibroblasts (MEFs). Indeed, the ATF4 transcriptional signature is the top gene set predicted to be upregulated upon combinatorial inhibition of MDM2 and PPM1D in our experiments (**Figure 1i**). Therefore, the answer to the Reviewer question is yes!, dual inhibition of the two p53 repressors upregulates genes that have been identified in other settings as part of the transcriptional response elicited by ISR activation. Inspired by the Reviewer's comment, we make this point very clear in the revised manuscript, including a new panel in **Figure S1i** showing the ATF4 target

genes driving the predicted activation of ATF4 by Ingenuity Pathway Analysis, while also citing the paper by Han et al.

6. Does the observed apoptosis depend on CHOP, a downstream target of the integrated stress response, or apoptosis inducers downstream of p53? This would address the broader question of whether ISR or p53-dependent cell death which is activated.

Response: We are grateful for this comment. As the Reviewer appreciates, both ATF4 and CHOP act downstream of ISR activation. Indeed, we show now that CHOP is also induced at the protein level upon dual treatment with MDM2 and PPM1D inhibitors (**Figure S3e**) and that knockdown of CHOP with two independent shRNAs impairs apoptosis in this setting (**Figure S3f**). These new results are described in the revised Results section.

7. Is it possible to rescue the heme reduction to show that this is truly underlying HRI activation? For example, by example silencing expression of HMOX1?

Response: We are grateful for this comment, Reviewer 1 also commented on this. Accordingly, we completed knockdown of HMOX1, which showed that, expectedly, reducing HMOX1 expression leads to increase in cellular heme levels and rescues the HRI-dependent apoptotic phenotype. These results are shown in new **Figure 3j-k**.

8. The use of the MTT assay could be problematic as it measures metabolic activity which may also be affected by the treatments. Therefore, MTT assay may not reflect cell viability.

Response: We agree with the caveats raised about the MTT assays, that is why we completed complementary measurements of apoptosis to substantiate the main conclusions. Prompted by the Reviewer's comment, we revised the Results text to clearly indicate that MTT assays measure metabolic activity.

9. There is earlier literature suggesting that apoptosis following treatment with p53 reactivation drugs requires activation of the integrated stress response (Johannes Ristau, Cell Death and Disease 2019 & Jun Yang, Mol Cell Biol 2009). This could be included in the discussion as it may indicate that this is a general feature.

Response: Thank you, we have incorporated these reports in the revised Discussion.

Minor concerns:

1. Figure 1i. The format "X versus Y" would normally mean that fold changes would be calculated via X-Y (in log scale). Here it seems to be the other way around.

Response: We agree with this comment and have changed the figure labels accordingly.

2. The authors occasionally use "protein translation" but proteins are not translated. The correct term is "mRNA translation" or "protein synthesis".

Response: We agree this comment and have used 'mRNA translation' in the revised manuscript.

Reviewer 3 (Remarks to the Author):

In this manuscript, the authors found that dual inhibition of MDM2 and eIF2A induces apoptosis in vitro and decreases tumor growth in mouse xenograft models.

The study is well-executed and thoroughly elucidates the combined roles of the p53 and eIF2A

pathways in apoptosis. However, the connection between PPM1D inhibition with GSK2830371 on ATF4 protein accumulation through eIF2A phosphorylation could be further developed. When MDM2 is inhibited and ATF4 is induced through eIF2A phosphorylation with GSK2830371 or nelfinavir, there is clear evidence for increased apoptosis. This is suggesting a greater role of ATF4 induction in this phenotype as opposed to PPM1D inhibition. The subject of the manuscript could be adjusted accordingly to acknowledge this. Additionally, the following suggestions will improve the manuscript before acceptance in Nat Comm.

Response: Thanks for these comments. Following Reviewer's guidance, in the revised text we highlight more prominently the role of ATF4 induction in the key phenotype observed.

Major comments:

1. Because PPM1D has been previously shown to dephosphorylate p53 at Ser15, perform a Western blot for Phospho-p53 (Ser15) after PPM1D inhibition with GSK2830371 as in Fig. 1A.

Response: Thanks for this comment. We have completed the S15P-p53 western blot, which expectedly shows increased S15 phosphorylation upon PPM1D inhibition. These new results are shown in **Figure 1a**.

2. To support the claim that ATF4 can undergo selective translation, measure ATF4 mRNA association with collected polysome fractions in Fig. 3C and 4C.

Response: Thanks for this comment, which was also raised by Reviewer 2. Expectedly, ATF4 mRNA shifts toward heavier polysome fractions upon dual inhibition of p53 repressors, while the control mRNA GAPDH shifts toward lighter fractions, consistent with the well demonstrated selective translation of ATF4 upon ISR activation¹³. This is now shown in new **Figure S4a**.

3. There are inconsistencies in the treatment times between similar experiments, making it difficult to make comparisons. Address the significant decrease in ATF4 mRNA expression after Nutlin treatment for 24 hours (Fig. S2A-B). Provide RT-qPCR analysis for ATF4 mRNA expression after 6 hours of Nutlin treatment as in Fig. S2A and a Western blot for ATF4 as in Fig. S2B after 24 hours of Nutlin treatment.

Response: We have completed the additional time points as requested by the Reviewer and included them in the revised manuscript (**Figures S2b, d**).

4. Fig 2b: A panel with ATF4 ChIP on ATF3 gene is required to show how ATF4 regulates ATF3 expression.

Response: We have completed the ATF4 ChIP experiment as requested (also requested by Reviewer 1), demonstrating binding of ATF4 to its response element (RE) in the ATF3 locus, as well as binding to the REs in BBC3/PUMA and GDF15 gene loci. New results are presented in **Figure 2j**.

5. Fig 2i: Although there is a trend for upregulation of p53 target genes after ATF4 overexpression, it does not seem to be significant based on the error bars. They should either reanalyze their results or increase their number of repeats to decrease the error bars.

Response: Thanks for the comment. In this experiment we reported the analysis of multiple p53 target genes, most of which are significantly more upregulated by Nutlin in cells overexpressing ATF4. This increased fold induction reaches statistical significance ($p < 0.05$) as defined by paired t test, where the fold induction in controls is compared to the fold induction in ATF4-overexpressing cells **within the same biological replicate experiment**. What the Reviewer is noting (overlapping error bars in some cases) is due to *variable induction upon Nutlin treatment* from one biological replicate to another. To

clarify this point, we re-plotted these data highlighting the paired nature of the three biological replicates (**Figure 2i, Figure S3a**). To further increase transparency of our data we replaced all other bar charts in the manuscript with one-dimensional scatter plots as suggested by Nature Communication editorial policy.

6. Fig S3b: How does EIF2AK1 mRNA levels go down after Nutlin+GSK administration while the protein levels go up (Figure 3b)? The authors should explain this unexpected result

Response: The Reviewer is correct that whereas HRI protein levels are elevated, HRI mRNA levels are downregulated. This phenomenon could be explained by either increased mRNA translation or increased protein stability. Analysis of polysome fractions did not reveal a clear change in polysomal distribution of the HRI mRNA (**Figure S4d**), suggesting potential protein stabilization, which has been previously documented for HRI¹⁴.

7. Fig S3f-g: The authors claim that the mitochondrial membrane potential is not changing but the graphs show the opposite. Are the changes not statistically significant?

Response: This comment prompted us to better display the results of this assay. In the original version of this figure we plotted total mean fluorescence intensity (MFI) values, which can be somewhat misleading, as they are impacted by cell size. For example, Nutlin treatment increases cell size and MFI. In the revised figure, we show both the actual distribution plots of MFI and bar plots displaying the % of TMRE positive cells (no collapse of mitochondrial membrane potential), which do not change significantly with the treatment. Updated panels are part of the **Figure S5**.

8. Fig 4e: What happens to ATF3 protein levels after Nutlin and Nelfinavir treatment? Is it the same as GSK+Nutlin?

Response: We now show a western blot for ATF3 in **Figure S6a**, demonstrating ATF3 accumulation with the combinatorial treatment.

Minor comments:

1. Some references to figure panels are missing from the text.

Response: Thank you, we have amended this oversight.

2. Statistics for Fig. S3D and S3F are missing.

Response: Thank you, we have re-plotted the original Figure S3f (now **Figure S5d**) and have added the statistics in former Figure S3d, now **Figure S5a**.

Bibliography.

- 1 Kleiblova, P. *et al.* Gain-of-function mutations of PPM1D/Wip1 impair the p53-dependent G1 checkpoint. *The Journal of cell biology* **201**, 511-521, doi:10.1083/jcb.201210031 (2013).
- 2 Zhang, L. *et al.* Exome sequencing identifies somatic gain-of-function PPM1D mutations in brainstem gliomas. *Nature genetics* **46**, 726-730, doi:10.1038/ng.2995 (2014).
- 3 Kahn, J. D. *et al.* PPM1D-truncating mutations confer resistance to chemotherapy and sensitivity to PPM1D inhibition in hematopoietic cells. *Blood* **132**, 1095-1105, doi:10.1182/blood-2018-05-850339 (2018).

- 4 Burocchiova, M. *et al.* Truncated PPM1D impairs stem cell response to genotoxic stress and promotes growth of APC-deficient tumors in the mouse colon. *Cell Death Dis* **10**, 818, doi:10.1038/s41419-019-2057-4 (2019).
- 5 Deng, W. *et al.* The role of PPM1D in cancer and advances in studies of its inhibitors. *Biomed Pharmacother* **125**, 109956, doi:10.1016/j.biopha.2020.109956 (2020).
- 6 Ianevski, A., Giri, A. K. & Aittokallio, T. SynergyFinder 2.0: visual analytics of multi-drug combination synergies. *Nucleic Acids Res* **48**, W488-W493, doi:10.1093/nar/gkaa216 (2020).
- 7 Zaccara, S. *et al.* p53-directed translational control can shape and expand the universe of p53 target genes. *Cell death and differentiation* **21**, 1522-1534, doi:10.1038/cdd.2014.79 (2014).
- 8 Andrysk, Z. *et al.* Identification of a core TP53 transcriptional program with highly distributed tumor suppressive activity. *Genome research* **27**, 1645-1657, doi:10.1101/gr.220533.117 (2017).
- 9 Rizzotto, D. *et al.* Nutlin-Induced Apoptosis Is Specified by a Translation Program Regulated by PCBP2 and DHX30. *Cell Rep* **30**, 4355-4369 e4356, doi:10.1016/j.celrep.2020.03.011 (2020).
- 10 Hisaoka, M. *et al.* Preferential translation of p53 target genes. *RNA Biol* **19**, 437-452, doi:10.1080/15476286.2022.2048562 (2022).
- 11 Nakano, K. & Vousden, K. H. PUMA, a novel proapoptotic gene, is induced by p53. *Mol Cell* **7**, 683-694, doi:10.1016/s1097-2765(01)00214-3 (2001).
- 12 Yu, J., Zhang, L., Hwang, P. M., Kinzler, K. W. & Vogelstein, B. PUMA induces the rapid apoptosis of colorectal cancer cells. *Mol Cell* **7**, 673-682 (2001).
- 13 Lu, P. D., Harding, H. P. & Ron, D. Translation reinitiation at alternative open reading frames regulates gene expression in an integrated stress response. *The Journal of cell biology* **167**, 27-33, doi:10.1083/jcb.200408003 (2004).
- 14 Alvarez-Castelao, B. *et al.* The switch-like expression of heme-regulated kinase 1 mediates neuronal proteostasis following proteasome inhibition. *Elife* **9**, doi:10.7554/eLife.52714 (2020).

REVIEWERS' COMMENTS

Reviewer #1 (Remarks to the Author):

The authors provide a substantially improved revised version of their manuscript. They have addressed all my concerns. I am happy to recommend the manuscript for publication in Nature Communications and I would like to congratulate the authors on this terrific study.

Reviewer #2 (Remarks to the Author):

Although the authors have not provided translome-wide data in response to my major point 3, the data presented as a reply (i.e. polysome-association of PUMA mRNA) do support the conclusions of the authors. For the data provided instead of translome data, I have one recommendation: The data presented in figure 4h (i.e. PUMA association with polysomes) are commonly presented as in figure S4A. I would recommend this presentation in figure 4h as well.s

My other concerns have been fully addressed.

Reviewer #3 (Remarks to the Author):

The authors have done a great job in addressing my comments.

Response to Referees Letter – NCOMMS-2200886.

Point by Point response to Reviewers' Comments.

Reviewer #1 (Remarks to the Author):

The authors provide a substantially improved revised version of their manuscript. They have addressed all my concerns. I am happy to recommend the manuscript for publication in Nature Communications and I would like to congratulate the authors on this terrific study.

Response: We are grateful to Reviewer #1 for the positive assessment.

Reviewer #2 (Remarks to the Author):

Although the authors have not provided translome-wide data in response to my major point 3, the data presented as a reply (i.e. polysome-association of PUMA mRNA) do support the conclusions of the authors. For the data provided instead of translome data, I have one recommendation: The data presented in figure 4h (i.e. PUMA association with polysomes) are commonly presented as in figure S4A. I would recommend this presentation in figure 4h as well.

My other concerns have been fully addressed.

Response: We are grateful for the positive assessment. Regarding representation of data in Fig. 4h, we would like to note that in this experiment we pooled the fractions of the polysome gradient into **light**, **medium**, and **heavy** subsets to detect the low abundance PUMA mRNA, so the data can not be presented exactly as the full gradient shown in Supplementary Fig. 4. Furthermore, the way we are presenting the data now in Fig. 4h highlights induction of polysomal PUMA mRNA in six experimental conditions as well as its redistribution from light to heavy fractions.

Reviewer #3 (Remarks to the Author):

The authors have done a great job in addressing my comments.

Response: We are grateful to Reviewer #3 for the positive assessment.